# Immunopotentiating Activity of Fucoidans and Relevance to Cancer Immunotherapy

**DOI:** 10.3390/md21020128

**Published:** 2023-02-15

**Authors:** Yani Li, Eileen McGowan, Size Chen, Jerran Santos, Haibin Yin, Yiguang Lin

**Affiliations:** 1School of Life Science, Faculty of Science, University of Technology Sydney, P.O. Box 123, Broadway, NSW 2007, Australia; 2Department of Immuno-Oncology, First Affiliated Hospital of Guangdong Pharmaceutical University, Guangzhou 510080, China; 3Guangdong Provincial Engineering Research Center for Esophageal Cancer Precision Therapy, Guangdong Pharmaceutical University, Guangzhou 510080, China; 4Key Laboratory of Cancer Immunotherapy of Guangdong High Education Institutes, Guangdong Pharmaceutical University, Guangzhou 510080, China; 5Guangdong Provincial Key Laboratory of CAR-T Cell Therapy Associated Adverse Effect Monitoring, Guangdong Pharmaceutical University, Guangzhou 510080, China; 6Advanced Tissue Engineering and Stem Cell Biology Group, Faculty of Science, University of Technology Sydney, P.O. Box 123, Broadway, NSW 2007, Australia; 7Research & Development Division, Guangzhou Anjie Biomedical Technology Co., Limited, Guangzhou 510535, China

**Keywords:** fucoidan, seaweed, immunopotentiation, immunomodulatory, immunotherapy, cancer treatment, cytokines

## Abstract

Fucoidans, discovered in 1913, are fucose-rich sulfated polysaccharides extracted mainly from brown seaweed. These versatile and nontoxic marine-origin heteropolysaccharides have a wide range of favorable biological activities, including antitumor, immunomodulatory, antiviral, antithrombotic, anticoagulant, antithrombotic, antioxidant, and lipid-lowering activities. In the early 1980s, fucoidans were first recognized for their role in supporting the immune response and later, in the 1990s, their effects on immune potentiation began to emerge. In recent years, the understanding of the immunomodulatory effects of fucoidan has expanded significantly. The ability of fucoidan(s) to activate CTL-mediated cytotoxicity against cancer cells, strong antitumor property, and robust safety profile make fucoidans desirable for effective cancer immunotherapy. This review focusses on current progress and understanding of the immunopotentiation activity of various fucoidans, emphasizing their relevance to cancer immunotherapy. Here, we will discuss the action of fucoidans in different immune cells and review how fucoidans can be used as adjuvants in conjunction with immunotherapeutic products to improve cancer treatment and clinical outcome. Some key rationales for the possible combination of fucoidans with immunotherapy will be discussed. An update is provided on human clinical studies and available registered cancer clinical trials using fucoidans while highlighting future prospects and challenges.

## 1. Introduction

Fucoidans are proving to be promising pharmaceutical agents, widely cited as exhibiting antitumor, immunomodulatory (immunopotentiating), anti-inflammatory, and other pharmacological properties [1,2,3,4,5,6,7,8]. These extracts contain exceedingly large amounts of vitamins and minerals, and, therefore, for many years, they have been used as safe dietary supplements to support and improve human immunity [9]. Fucoidan extracts, as soluble dietary fibers, play critical roles in providing nourishment with therapeutic benefits in the prevention of disease and cancer treatment [3,9]. Due to the medicinal benefits of fucoidans, they have been used successfully as a natural source of iodine to treat thyroid complaints in several Asian and Western countries [10]. Although registered as a Food and Drug Administration (FDA)-approved safe food supplement with multiple beneficial effects, the FDA has not approved therapeutic fucoidan products for clinical use. The underlying problems and obstacles associated with the clinical use of fucoidan(s) is discussed in this review, with particular focus on the immunopotentiating activity of fucoidans and their relevance to cancer immunotherapy.

Fucoidans were first isolated from the extracellular mucus matrix of brown macroalgae, class Phaeophycceae, species *Fucus vesiculosus*, *Asophyllum nodosum*, *Laminaria digiata*, and *Laminaria saccharina*, in 1913, by the Swedish professor Hareld Kylin at Uppsala University [11,12,13]. The name fucoidan was initially used as a general term referring to a variety of high-molecular weight sulfated polysaccharides derived from fibrillar cell walls and mucous matrices of diverse species of brown macroalgae [14]. Until relatively recently, the International Union of Pure and Applied Chemistry (IUPAC) based the fucoidan nomenclature on the generic sulfated fucan structure—a polysaccharide with a backbone structure based on sulfated L-fucose residues, with the main structure consisting of a minimum of 10 monosaccharides [15]. Hence, the names fucoidan, fucoidans, and sulfated fucans have been used synonymously in the literature, distinct from fucans and fucosans. Recently, in 2017, the term fucoidan(s) from brown algae was revised, and now is clearly established to refer to heteropolymers with more diverse backbones [14].

The bioactivities of these versatile marine-origin heteropolysaccharides differ depending on the seaweed species and environmental growth conditions. There are approximately 250 genera and 1500–2000 species of brown algae, and chemical composition, structure of fucoidans, and bioactivity differ from species to species [16]. Fucoidan extracts from various species of brown seaweed consist of different amounts of monosaccharide compositions, including fucose, mannose, galactose, glucose, and xylose [17]. The proportion of these monosaccharides within each fucoidan influences its capability to induce a particular pharmacological activity. For example, *Fucus vesiculosus* fucoidan shows a strong 1,1-diphenyl-2-picrylhydrazyl (DPPH) radical scavenging activity [17] and *Ascophyllum nodosum* fucoidan promotes T cell proliferation [18]. Hence, it is important to be aware that the source of a fucoidan plays a critical role in inducing its pharmacological activities, and, therefore, throughout this review we have identified the source, molecular weight of the fucoidan, and subsequent fraction isolates when it has been specified in the literature. Methods of extraction are also major factors affecting the structural composition and bioactive properties of fucoidans both within and between brown algae species (discussed in Section 2).

All these factors mentioned contribute to the difficulty in determining the type of fucoidan and the effective dose of a specific fucoidan for use in preclinical and clinical studies. Fucoidan derivatives or variants, where the molecular weight of fucoidan has been substantially reduced to as little as two or three monosaccharides, have been shown to lack the immune ability (activation) of a high molecular weight fucoidan (HMWF) [19]. Highly purified fucoidan fractions, defined as low molecular weight fucoidans (LMWFs), also fall below the standard IUPAC definition of a fucoidan with reduced or altered bioactivity. Hence, the molecular weight and bioactivity of fucoidans and their variants used in preclinical studies influence efficacy and are important discussion points in this review.

Cancer immunotherapy, including immune checkpoint inhibitors (ICI) and adoptive T cell transfer therapy (such as CAR-T cell therapy), has revolutionized cancer treatment and achieved an unprecedented clinical outcome in the treatment of homological malignancies. However, significant challenges remain ahead in the treatment of solid tumors using these approaches. Effective treatment of solid tumor requires preferential activation of CTL-mediated cytotoxicity against cancer cells. To overcome these seemingly intractable obstacles, more ‘powerful’ immune cells or CAR-T cells with enhanced antitumor efficacy are required [20]. To improve ICI efficacy, combinations with additional therapeutics, such as fucoidans, may be a viable approach [18]. Fucoidans are natural polysaccharides that have antitumor properties and immunopotentiating effects with low toxicity in both animal models and humans. These polysaccharides are strong agonists of TLR4, which induces the activation of DCs in humans. Therefore, fucoidans could be desirable candidates for use in enhancing cancer immunotherapy [21].

Although fucoidans are promising antitumor candidates in cancer therapy, their efficacy will be highly dependent on the species type and the extraction methods of the fucoidan compounds. Fucoidans as adjuvants have been found to enhance the efficacy of ICIs in the treatment of melanoma [18,22] and metastatic lung cancer [23]. As such, fucoidans, used in combination with other cancer immunotherapy drugs, have made significant progress in the field of cancer. In clinical studies, fucoidan (*Undaria pinnatifida*), co-administered with 20 mg of tamoxifen to treat patients with breast cancer, demonstrated increased treatment efficacy and adverse effects were not detected (ACTRN12615000673549) [24], suggesting that fucoidans could be co-administered with other hormonal therapeutic drugs and act as efficacious adjuvants. However, detailed immunomodulatory mechanisms of fucoidans are elusive and have not been fully elucidated. Hence, this review aims to provide comprehensive information on the immunomodulatory mechanisms of fucoidans, including the association between fucoidan and inflammatory cytokines and signaling pathways to emphasize the potential of fucoidans in modern cancer immunotherapies. Finally, we discuss current clinical studies and the clinical efficacy of fucoidans in cancer treatments, especially focusing on cancer immunotherapy.

## 2. Overview of Fucoidans

### 2.1. Structural Characteristics of Fucoidans

The structure and composition of fucoidans vary depending on the source [25], marine species [8,14,25,26,27,28,29,30], harvest season [6,31], and extraction methods (Table 1) [6,29,31]. When harvested from various sources and species at different times of the year, fucoidans have been shown to have distinct structural characteristics [14,25,26,27,28,29,30], and the compositional properties are often varied when derived using different extraction methods, which have been extensively reviewed in several species [14,25,26,27,28,29,30]. In particular, extraction methods and sources are the two critical factors that influence the proportions of monosaccharides contained within different fucoidan extracts (Table 1) [32,33,34], leading to the inducement of different immune activities. For example, previous studies have shown that fucoidan extracts from *Ecklonia cava*, *Ascophyllum nodosum*, *Undaria pinnatifida, Laminaria japonica*, and *Fucus vesiculosus* induced T cell activation [23], promoted T cell proliferation [18], dendritic cell (DC) maturation [35], macrophages [36], and activation of natural killer cells (NK), respectively. Although fucoidans have no universal structure, the fucoidan extracted from *Fucus vesiculosus*, known as bladderwrack, has the most straightforward and typical fucoidan chemical structure among all species of brown seaweed. Patankar et al. (1993) reported that fucoidan derived from *Fucus vesiculosus* has a base of repeating α(1→3)-linked *α*-L-fucopyranose units with a substituted sulfate group at the C-4 position on the chain and fucose enabling branching points within the chain [29,37,38]. Subsequently, two additional structural models of fucoidan have been reported: (1) an alternated (1→4)-linked *α*-L-fucopyranose units [29,34,39,40]; (2) an alternated (1→3) and α(1→4)-linked sulfated L-fucopyranose [4,29,40,41,42]. As previously mentioned, fucoidans are heteropolysaccharides that consist primarily of abundant L-fucose residues and sulfate groups, but they also contain traces of monomers, such as galactose, uronic acid, xylose, mannose, glucose, and glucuronic acid [12,29,39,41,42,43,44]. Previous characterization demonstrated variations in monosaccharide composition, molecular weight, types of glycosidic linkages, the presence of branching, and the degree of sulfation [32,43,45,46]. The importance of understanding these structural and compositional variations is that they can have varying impacts on therapeutic effects in cancer treatment and other inflammatory-based diseases.

The pharmacological effects of fucoidan vary with their molecular weight. Although the IUPAC definition of fucoidan is a minimum of 10 monosaccharides, fractional derivatives are commonly used to assess fucoidan bioactivity [47]. Fucoidans are usually classified as being either a low molecular weight fucoidan (LMWF) or high molecular weight fucoidan (HMWF), as the pharmacological effects of fucoidans vary with their molecular weight. Generally, LMWF refers to molecular weight < 10 kDa, while HMWF refers to a fucoidan with a molecular weight > 10,000 kDa, and medium molecular weight is 10–10,000 kDa [33,48]. However, other standards of classification are also used. For example, HMWF means a fucoidan with an average molecular weight of 300 kDa or more, while a fucoidan with an average MW < 300 kDa is considered LMWF [49].

**Table 1 marinedrugs-21-00128-t001:** Fucoidans from various algae.

Name of Algae	Place of Collection	Extraction Method	Yield (%)	Monosaccharide Composition (%)	Molecular Weight (kDa)	Ref.
*Ascophyllum nodosum*	ND	Acid hydrolysis, centrifugal partition, chromatography.	ND	Fuc: ND.	ND.	[50]
*Cladosiphon navae-caledoniae*	Daiichi Sangyo Corporation (Osaka, Japan)	ND	HMWF: 85, LMWF: 72	ND LMWF: Fuc: 73, xyl: 12, man: 7	LMWF: <500.	[33]
*Fucus serratus*, *Fucus vesiculosus*, *Ascophyllum nodosum*	Coast of Aberystwyth at low tide, UK	CaCl_2_ extraction	6.0 wt, 9.8 wt, 8.0 wt	fuc:18–28 wt, sulphate: 30–40 wt; fuc: 26–39 wt, sulphate: 9–35 wt; fuc: 35–46 wt, sulphate: 6–22 wt.	1608; 1364 1374	[31]
*Fucus* *vesiculosus*	Purchased from Sigma	ND	ND	Fuc: 97, gal: ND, xyl: ND.	100	[37]
*Fucus* *vesiculosus*	Fucoidan extract 1 (FE1): from Marinova; FE2 and FE3 from Sigma-Aldrich	ND	FE1: 52.5 FE2: 52.2 FE3: 50.5	FE1: Fuc: 73.1, xyl: 8.0, man: 1.3, gal: 3.5, glu: 0.7. FE2: Fuc: 79.1 xyl: 3.9, man: 0.8, gal: 5.5, glu: 0.8. FE3: Fuc: 71.2, xyl: 5.3, man: 1.5, gal: 5.4, glu: 1.3.	FE1: 91 FE2: 60 FE3: 42	[32]
*Fucus* *evanescens*	Littoral of Iturup island (Kuril Islands)	CaCl_2_ extraction	Fraction F1: 3.9, Fraction F2: 2.6, Fraction F3: 21.4, Fraction F4: 47.4 Fraction F5: 4.5	F1: fuc: 35.4, xyl: 6.1, man: 0.8, glu: 4.0. F2: fuc: 10.7, xyl: 17.4, gal: 3.0, man: 3.7, glu: 1.1. F3: fuc: 33.2, xyl:8.1, gal:4.5, man: 3.5 F4: fuc: 58.7, xyl: 1.6, gal: 1.6. F5: fuc: 34.0, xyl: 3.8, gal: 5.4.	ND.	[51]
*Laminaria Saccharina*, *Laminaria digitata*, *Cladosiphon okamuranus*, *Fucus* *evanescens*, *Fucus* *vesiculosus*, *Fucus serratus*, *Fucus distichus*, *Fucus spiralis*, *Ascophyllum nodosum*	ND	CaCl_2_ extraction	ND	Fuc: 36.7 (*w*/*w*), xyl: 1.2 (*w*/*w*), man: 1.0 (*w*/*w*), glu: 2.2 (*w*/*w*), gal: 4.6 (*w*/*w*). Fucose: 30.1 (*w*/*w*), xyl: 1.9 (*w*/*w*), man: 1.7 (*w*/*w*), glu: 1.4 (*w*/*w*), gal: 6.3 (*w*/*w*). Fuc: 30.9 (*w*/*w*), xyl: 0.7 (*w*/*w*), glu: 2.2 (*w*/*w*). Fuc: 58.7 (*w*/*w*), xyl: 1.6 (*w*/*w*), gal: 1.6 (*w*/*w*). Fuc: 26.1 (*w*/*w*), xyl: 2.4 (*w*/*w*), man: 3.1 (*w*/*w*), glu: 2.2 (*w*/*w*), gal: 5.0 (*w*/*w*). Fuc: 24.8 (*w*/*w*), xyl: 2.4 (*w*/*w*), man: 2.1 (*w*/*w*), glu: 2.0 (*w*/*w*), gal: 4.8 (*w*/*w*). Fuc: 40.8 (*w*/*w*), xyl: 0.8 (*w*/*w*), gal: 0.8 (*w*/*w*). Fuc: 33.0 (*w*/*w*), xyl: 2.8 (*w*/*w*), man: 1.4 (*w*/*w*), glu: 1.2 (*w*/*w*), gal: 3.0 (*w*/*w*). Fuc: 26.6 (*w*/*w*), xyl: 4.4 (*w*/*w*), man: 2.6 (*w*/*w*), glu: 1.1 (*w*/*w*), gal: 4.7 (*w*/*w*).	200–500	[34]
*Laminaria hyperborea*	ND	ND	ND	Fuc: 97.8, gal: 2.2, glu: ND.	469.2	[45]
*Undaria* *pinnatifida*	Great Barrier Island, Port Underwood New Zealand	CaCl_2_ extraction	Increased from July (25.4–26.3) to September (57.3–69.9).	Crude fucoidan (F0): L-fuc: 39.24, D-gal: 26.48, D-xyl: 28.85, D-man: 5.04, α-D-glu: 0.95. Commercial fucoidan: L-fuc: 87.12, D-gal: 5.69, D-xyl: 4.85, D-man: 1.39, α-D-glu: 0.94. Fucoidan fraction F1: L-fuc: 48.51, D-gal: 37.86, D-xyl: 3.74, D-man: 6.97, α-D-glu: 2.91. F2: L-fuc: 53.21, D-gal: 42.12, D-xyl: 1.15, D-man: 2.24, α-D-glu: 1.28. F3: L-fuc: 59.71, D-gal: 28.74, D-xyl: 1.58, D-man: 7.19 α-D-glu: 2.77.	F0: 171 F1: 81 F2: 22 F3: 27	[6]
*Undaria* *pinnatifida*	Five fucoidans purchased from Sigma Aldrich, Anhui Minmetals Development I/E Co. Ltd., Matakana SuperFoods, Glycomix UK, and Leili Ltd	CaCl_2_ extraction	*Undaria pinnatifida* fucoidan (S): ≥95, Crude fucoidan (S1): ND, S2: 75.5 S3: ND, S4: 75 S5: ND	Fucoidan(S): fuc: 27.44, gal: 25.34, Fraction S1: fuc: 19.50, gal: 21.20. Fraction S2: glu: 96.71, Fraction S3: fuc: 13.83, gal: 13.24. Fraction S4: fuc: 20.35, gal: 19.26. Fraction S5: fuc: 19.23, gal: 21.00, glu: 6.38.	S: 440 S1: 440–2000 S2: <2000 S3: 38.9–440 S4: 35.2–440 S5: >2	[52]
*Undaria* *Pinnatifida*	From Sigma Aldrich	CaCl_2_ extraction	*Undaria pinnatifida* fucoidan: ≥95 LMWF: ND	ND.	ND LMWF: <10	[48]
*Undaria* *Pinnatifida* *Fucus* *vesiculosus*	Port Underwood, New Zealand	CaCl_2_ extraction	Sporophyll derived from farm 327: 69, Sporophyll derived from farm 106: 57.28	Crude fucoidan (F0): fuc: 39.24, xyl: 28.85, gal: 26.48, man: 5.04, glu: 0.95, F1: fuc: 48.51, xyl: 3.74, gal: 37.86, man: 6.97, glu: 2.91. F2: fuc: 53.21, xyl: 1.15, gal: 42.12, man: 2.24, glu: 1.28. Fraction F: fuc: 59.71, xyl: 1.58, gal: 28.74, man: 7.19, glu: 2.77, Fucoidan (Sigma): fuc: 87.12, xyl: 4.85, gal: 5.69, man: 1.39, glu: 0.94.	F0: >150 F1: 81, F2: 22 F: 27 54	[53]

Abbreviations are as follows: ND, not detailed; HMWF, high molecular weight fucoidan; LMWF, low molecular weight fucoidan; gal, galactose; fuc, fucose; xyl, xylose; man, mannose; glu, glucose.

### 2.2. Pharmacological Actions of Fucoidans

Fucoidans have been shown to exhibit a variety of beneficial pharmacological effects, including antitumor [5,54,55,56,57,58], anti-inflammatory [3,4,5], immunomodulatory [4,5] antioxidant [6,7,8], anticoagulant [5,59,60,61,62], antithrombotic [5,63,64], antiangiogenic [5,56,65,66,67,68,69], and antiviral [70,71,72,73,74,75].

The antitumor effects of fucoidans have been extensively investigated in vitro in various tumor cell lines, especially in lung and breast cancer cell lines [1,76,77], and in vivo in animal models [78]. The antitumor mechanisms of fucoidans in these tumor cells (e.g., A549, MCF-7) include cell cycle arrest at the sub-$G_{1}$/$G_{1}$ phase [79,80,81,82], caspase-dependent apoptosis [80,81,83], regulation of specific apoptotic proteins (e.g., poly [ADP-ribose] polymerase 1 (PARP1) [84,85,86], protein kinase RNA-like endoplasmic reticulum kinase (PERK), B-cell lymphoma 2 (Bcl-2) [84], BAX [84,87], and caspases -3, -8, and -9 [80,84,86]). Increasing evidence has shown that fucoidans are capable of exhibiting direct and indirect inhibitory effects on tumor cells by regulating several important signaling pathways, such as extracellular signal-regulated kinase 1/2 (ERK1/2) [56,84,88], phosphoinositide 3-kinase-Akt (PI3K/Akt) [84,88], p38 mitogen-activated protein kinase (p38 MAPK) [84], and mammalian target of rapamycin (mTOR) pathways [87,88].

In the in vitro study by Miyamoto et al. (2009) [80], they found that fucoidan (*Cladosiphon okamuranus*) was cytotoxic to MCF-7 cells and resulted in a significant increase in the number of apoptotic MCF-7 cell bodies, with condensation of chromatin and DNA fragmentation at the sub-$G_{1}$ phase of the cell cycle [80]. They then demonstrated that cell cycle arrest at the sub-$G_{1}$ phase was accompanied by activation of caspases (caspase 7, -8, -9) and PARP cleavage in fucoidan-treated MCF-7 cells at a concentration of 1000 μg/mL [80]. These results suggest that fucoidan (*Cladosiphon okamuranus*) induces apoptosis through activating specific caspases (e.g., caspase-7, -8, -9) and interacts with apoptotic proteins (e.g., PARP) to induce apoptosis in MCF-7 cells at a high concentration (e.g., 1000 μg/mL). In another in vitro study, Zhang et al. showed that fucoidan (Mozuku, *Cladosiphon novaecaledoniae*) caused an accumulation of apoptotic MCF-7 cells at the $G_{1}$ phase of the cell cycle [79], and these apoptotic MCF-7 cells presented with shrunken nuclei and fragmented chromatin [79]. They found that fucoidan (Mozuku, *Cladosiphon novaecaledoniae*) was cytotoxic to MCF-7 cells and inhibited 60% of MCF-7 cell growth at a concentration of 820 μg/mL [79]. Banafa et al. showed, in vitro, that fucoidan (*Fucus vesiculosus*) could induce apoptosis in MCF-7 cells through inducing cell cycle arrest at the $G_{1}$ phase by downregulating the expression levels of cyclin D1 and CDK-4 in MCF-7 cells [83]. They also revealed that fucoidan (*Fucus vesiculosus*) could induce apoptosis by downregulating the anti-apoptotic protein of Bcl-2 and upregulating the pro-apoptotic protein of Bax in fucoidan-treated MCF-7 cells [83]. In addition, they demonstrated that treatment with fucoidan (*Fucus vesiculosus*) could activate caspase-8 and increase cytochrome C release in MCF-7 cells [83], supporting the theory that a caspase-dependent pathway may lead to apoptotic protein cleavage, such as Bid cleavage [83]. Furthermore, a more recent in vitro study by Abudabbus et al. (2017) showed that commercial fucoidan (*Fucus vesiculosus*) caused cell cycle arrest at the sub-$G_{1}$/$G_{1}$ phase with activation of caspases 3, -7, and -9 [81]. Overall, these results suggest that the fucoidan (e.g., *Cladosiphon novaecaledoniae*, *Fucus vesiculosus*)-induced apoptosis involves the activation of caspases (caspase 3, -7, -8, -9) and cell cycle arrest at ${\mathrm{the}\;G}_{0}/G_{1}$ phase in MCF-7 cells.

The molecular weight of fucoidan also influences its potential to induce apoptosis in MCF-7 and MDA-MB-231 cell lines. In a recent in vitro study conducted by Lu et al. (2018), they demonstrated that a LMWF from New Zealand *Undaria pinnatifida* was cytotoxic and exhibited maximum anti-proliferative effects in both MCF-7 and MDA-MB-231 cells at concentrations of 200 μg/mL and 300 μg/mL, respectively [52]. They also demonstrated that this LMWF (New Zealand *Undaria pinnatifida*) significantly increased the total caspase in MDA-MB-231 cells at a concentration of 300 μg/mL [52]. Notably, the concentration of fucoidan used in the study was much lower than in the previous studies by Zhang et al. (2013) [89], Miyamoto et al. (2009) [80], and Banafa et al. (2013) [83]. Therefore, these results suggest that the effective concentration of fucoidan used to induce apoptosis in breast cancer cell lines (MCF-7, MDA-MB-231) varies significantly.

Regardless of the species of fucoidan, all fucoidans tested inhibited A549 cell growth and proliferation by adjusting their concentrations in both in vitro and in vivo studies. An in vitro study conducted by Boo et al. (2011) showed that fucoidan (*Undaria pinnatifida*) significantly inhibited A549 cell growth leading to the accumulation of apoptotic A549 cell bodies at the sub-$G_{1}$ phase of the cell cycle at a concentration of 200 μg/mL [84]. These apoptotic A549 cell bodies presented with chromatin condensation, membrane blebbing, and cell shrinkage [84]. Boo et al. (2011) also demonstrated that fucoidan (*Undaria pinnatifida*) could induce apoptosis by activating caspase-9, decreasing pro-caspase-3, and then inducing PARP cleavage in A549 cells at a concentration of 200 μg/mL [84]. Furthermore, they revealed that fucoidan-induced A549 cell apoptosis is also associated with downregulation of antiapoptotic Bcl-2 proteins and upregulation of pro-apoptotic BAX proteins at a concentration of 200 μg/mL [84]. In addition, they also demonstrated that fucoidan (*Undaria pinnatifida*) activated the ERK1/2 signaling pathway [84], whilst inhibiting the p38 MAPK and PI3K/Akt signaling pathways [84], suggesting that fucoidan (*Undaria pinnatifida*) induces apoptosis in A549 cells through regulating the MAPK-based signaling pathways (ERK1/2, p38 MAPK, and PI3K/Akt). In particular, this was the first in vitro study in which researchers confirmed that fucoidan-induced apoptosis is associated with the regulation of apoptotic proteins (e.g., PARP, Bcl-2, Bax), signaling pathways (e.g., ERK1/2, p38 MAPK, PI3K/Akt), and specific caspases (e.g., caspase-9, pro-caspase-3) in A549 cells [84]. Similarly, an in vitro study by Hsu et al. (2017) demonstrated that commercial fucoidan (*Fucus vesiculosus*) induced cell cycle arrest at the sub-$G_{1}$/$G_{1}$ phases by upregulating p21 gene expressions in A549 cells at a concentration of 200 μg/mL and induced apoptosis by activating PARP proteins and caspase-3 in A549 cells at a concentration of 400 μg/mL [86]. They also reported that the commercial fucoidan (*Fucus vesiculosus*) could induce apoptosis through inducing an ER stress response by activating the PERK/ATF4/CHOP pathways [86]. These results suggest that fucoidan induces A549 cell apoptosis not only through regulating apoptotic proteins (e.g., PARP, Bcl-2, Bax), genes (e.g., p21) and caspases (e.g., caspase-9 and -3), but also by regulating extrinsic signaling pathways, such as the PERK/ATF4/CHOP pathways.

Furthermore, similar results were also obtained in in vitro studies by Lee et al. (2012) and Hsu et al. (2018) [85]. Lee et al. (2012) demonstrated that commercial fucoidan (*Fucus vesiculosus*) inhibited cell proliferation of A549 cells at a concentration of 400 μg/mL [88], and Hsu et al. (2018) demonstrated that fucoidan treatment could significantly inhibit A549 cells at concentrations of 200 and 400 μg/mL (${\mathrm{IC}}_{50}$ = 10 μM) [85]. Lee et al. (2012) found that commercial fucoidan (*Fucus vesiculosus*) could suppress MMP-2 activity, migration, and invasion of A549 cells at a concentration of 200 μg/mL, and also revealed that commercial fucoidan (*Fucus vesiculosus*) could inhibit MMP-2 expression, metastasis, and invasion by inhibiting phosphorylation levels of the ERK1/2 and PI3K-Akt-mTOR pathways and its downstream targets 4E-BP1 and p70S6K at a concentration of 200 μg/mL [88]. Thus, these results suggest that fucoidan inhibits the MMP-2 activity by regulating the ERK1/2 and/or PI3K-Akt pathways but may only partially involve the mTOR pathway. These results also indirectly indicate that commercial fucoidan (*Fucus vesiculosus*) inhibits A549 cell growth and proliferation by downregulating the PI3K-Akt-mTOR and ERK1/2 signaling pathways in a concentration range of 200–400 μg/mL. Furthermore, the in vitro study by Chen et al. (2021) also demonstrated that fucoidan inhibits A549 cell proliferation by decreasing the level of protein expression of p-mTOR and downstream proteins p-S6K, p-P70S6K, and p-4EBP1 in the mTOR pathway [87]. They also demonstrated that fucoidan induces apoptosis by reducing the expression levels of anti-apoptotic Bcl-2 proteins and increasing the expression levels of pro-apoptotic BAX proteins in A549 cells [87]. These results reinforce that fucoidan-induced apoptosis in A549 cells is associated with the balance of the Bcl-2/Bax ratio, and the regulation of the ERK1/2 and PI3K-Akt-mTOR pathways. Taken together, these results substantiate the theory that the balance of the apoptotic protein ratio and the activation of signaling pathways are essential for fucoidan to induce apoptosis in A549 tumor cells.

More importantly, the in vivo study by Chen et al. (2021) also demonstrated that fucoidan (*unknown source*) significantly inhibits the volumes and weights of A549 cells in xenograft mice after oral intake of fucoidan (25 mg/kg) for 14 days [87], and the expression value of Ki67 was also lower in the fucoidan-fed group than in the control group [87]. These results are consistent with a previous in vivo study conducted by Chen et al. (2016). This study also showed that fucoidan reduced approximately 23.3% and 40.3% of the mean tumor volume and weight at 0.5 mg/kg and 10 mg/kg, respectively, in the tumor xenograft model A549 [90], and also confirmed that the expression value of Ki67 was lower than in the control group [90]. These results support the theory that fucoidan directly induces A549 cell apoptosis in an in vivo model. The anti-inflammatory and immunomodulatory effects of fucoidan are interlinked with the antitumor effects. An overview of these effects is presented here, and a more detailed discussion is given in later sections. Upon exposure to pathogen-associated molecular patterns (PAMP), viruses and bacteria, innate (e.g., macrophages, dendritic cells (DC), natural killer (NK) cells), and adaptive immune cells (e.g., CD4+ helper and CD8+ cytotoxic T cells), are activated in response to inflammation [2,3]. The development of lung and breast carcinomas is associated with the activation of some vital pro-inflammatory biomarkers, including cytokines (e.g., TNF-*α*, IL-1, IL-6, IL-12), chemokines, cyclooxygenase-2 (COX-2), prostaglandins, C-reactive protein (CRP) [91], inducible nitric oxide synthase (iNOS) and nitric oxide (NO), which facilitate tumor initiation and progression [3,92]. Many previous studies have reported that fucoidan regulates these biomarkers and MAPK signaling pathways to induce anti-inflammatory and immunomodulatory effects in immune cells [18,23,93,94,95,96,97]. Furthermore, the upregulated expression of these biomarkers activates intrinsic and extrinsic signaling pathways, such as the NF-κB and MAPK pathways [3,98]. Therefore, regulating the expression of these pro-inflammatory biomarkers is critical to overall disease control and management.

Some studies demonstrating the anti-inflammatory effects of fucoidans are presented here for discussion. In an in vitro study, Jeong et al. (2017) reported that fucoidan (*Fucus vesiculosus*) was non-cytotoxic to RAW264.7 murine macrophages at 100 μg/mL and attenuated the production of pro-inflammatory cytokines of TNF-*α* and IL-1*β* in RAW264.7 murine macrophages treated with liposaccharide (LPS) [96]. These results are consistent with the findings presented in a recent in vitro study where they showed that a particular fucoidan (*Saccharina japonica*) fraction (LJSF4) induced a strong inhibitory effect on the production of pro-inflammatory cytokines, including TNF-*α*, IL-1*β*, and IL-6 [97]. These results strengthen the argument that fucoidan, regardless of its species derivation, has the ability to induce anti-inflammatory effects in RAW264.7 macrophages. Furthermore, Lee et al. (2012) found that three *Ecklonia cava* fucoidan fractions ($F_{1}$, $F_{2}$, $F_{3}$) were not toxic to the RAW 264.7 macrophages at a concentration of 12.5–100 μg/mL and could significantly inhibit NO production, iNOS, COX-2 mRNA expression level, and pro-inflammatory cytokines (TNF-*α*, IL-1*β*, and IL-6) in LPS-treated RAW264.7 macrophages [99], where the $F_{3}$ fraction had the highest inhibitory effects on NO production [99]. These results suggest that there may be an association between the inhibition of the iNOS/NO pathway and attenuation of pro-inflammatory cytokines (TNF-*α*, IL-1*β*, and IL-6), but more studies are required to verify whether there is a connection between these two stimuli. Furthermore, the investigations by Fernando et al. (2017) also showed similar results; they found that the F2,4 fraction (*Chnoospora minima* fucoidan) was non-cytotoxic to LPS-treated RAW264.7 macrophages and exhibited a maximum inhibition in NO production, while the mRNA expression of pro-inflammatory cytokines (TNF-*α*, IL-1*β*, IL-6) was decreased in LPS-treated RAW264.7 macrophages at a concentration of 50 μg/mL [100]. Sanjeewa et al. (2018) reported that the *Sargassum horneri* fucoidan f4 fraction (36.86% fuc and 18.47% sulfate contents) exhibited the maximum inhibitory effects on NO production and significantly inhibited the mRNA expression levels of the pro-inflammatory cytokines of TNF-*α* and the ${\mathrm{PEG}}_{2}$ enzyme in LPS-treated RAW264.7 murine macrophages without causing any toxicities [101], but only slightly inhibited IL-6 production [101]. They also demonstrated that the f4 fraction could dose-dependently inhibit the expression of the iNOS and COX-2 proteins [101], as well as inhibit the phosphorylation of IκB-α and p-I κB-α [101]. These results suggest that the *Sargassum horneri* fucoidan f4 fraction induces anti-inflammatory effects in LPS-treated RAW264.7 macrophages through the NF-κB pathway [101].

Sanjeewa et al. (2017) demonstrated that a crude fucoidan extracted from *Sargassum horneri* had no cytotoxicity to RAW264.7 murine macrophages and exhibited a maximum suppressive effect on NO production in LPS-treated RAW264.7 murine macrophages [102]; it also reduced the secretion of pro-inflammatory cytokines TNF-*α* and IL-1*β* and downregulated the expression levels of iNOS and COX-2 proteins in RAW264.7 murine macrophages treated with LPS [102]. More importantly, they also showed that CCP fucoidan (*Sargassum horneri*) (100 μg/mL) induced anti-inflammatory effects in RAW264.7 murine macrophages treated with LPS by inhibiting the translocation of NF-*κ*B p50 and p65 to the nucleus and downregulation of phosphorylation of p38 and ERK1/2 that was shown to increase with LPS stimulation in RAW264.7 macrophages [102].

Fucoidan has also been shown to act as a pro-inflammatory cytokine modulator in other types of cells, including mesenchymal stem cells (MSCs) [103], THP-1 monocytes [104], human peripheral blood mononuclear cells (PBMC) [105], porcine peripheral blood polymorphonuclear cells (PMN) [106], and human intestinal epithelial cells (Caco-2 cells) [107], indicating that fucoidan is capable of inducing anti-inflammatory effects in a broad range of cells. In an earlier in vitro study, Hwang et al. (2016) reported that LMWF (*Sargassum hemiphyllum*) induced anti-inflammatory effects in Caco-2 cells by downregulating LPS-induced TNF-α and IL-1β [107], which is in line with the findings from another in vitro study conducted by Ahmad et al. (2021) [105]; they demonstrated that fucoidan (*Undaria pinnatifida*) induced significant inhibitory effects on the pro-inflammatory cytokines of TNF-α and IL-6 in LPS-treated human PBMC cells [105]. However, the expression of IL-1β was only slightly reduced in this study [105], suggesting that fucoidan (*Undaria pinnatifida*) is less effective in reducing the expression of IL-1β in LPS-treated human PBMC cells as compared to the potency of the other two pro-inflammatory cytokines, namely TNF-α and IL-6. Furthermore, they also showed that fucoidan (*Fucus vesiculosus*) significantly inhibited three pro-inflammatory cytokines, namely TNF-α, IL-1β, and IL-6, in LPS-treated human PBMC [105]. Taken together, these results support the theory that *Fucus vesiculosus* fucoidan has a greater capability than *Undaria pinnatifida* fucoidan to exhibit inhibitory effects on pro-inflammatory cytokines, namely TNF-α, IL-1β, and IL-6, in LPS-treated human PBMC cells. It also indirectly indicates that *Fucus vesiculosus* fucoidan may have a greater cytotoxic level than *Undaria pinnatifida* fucoidan in LPS-treated human PBMC cells. *Macrocystis pyrifera* fucoidan and fractionated *Macrocystis pyrifera* fucoidan (5–30 kDa) also significantly inhibited pro-inflammatory cytokines TNF-*α* in LPS-treated THP-1 cells [105]. Furthermore, Kim et al. (2018) reported that fucoidan has a negative effect on inhibiting the production of TNF-α in PBMC [106]; this was found in the process of suppressing excessive phagocytosis of porcine peripheral blood PMN [106]. However, the mRNA expression level of TNF-α was reduced by adding fucoidan to LPS-stimulated PBMCs [106]. These results are in line with those reported in MSCs [103], they found that after coculture of MSCs treated with fucoidan with LPS-stimulated macrophages, the level of pro-inflammatory cytokines of TNF-α decreased, suggesting that MSCs treated with fucoidan are capable of inhibiting the production of TNF-α in LPS-stimulated macrophages. Another important in vitro study designed three fucoidan/chitosan nanoparticles for the topical delivery of methotrexate [104]. To assess the anti-inflammatory effect of this compound, Barbosa et al. (2019) characterized the fucoidan/chitosan (F/C) nanoparticles into three groups based on the weight ratio of fucoidan and chitosan, named 1F/1C, 3F/1C, and 5F/1C [104]. According to their findings, the 5F/1C nanoparticle contained a large amount of fucoidan and showed the highest inhibitory effects on the production of pro-inflammatory cytokines, namely TNF-α, IL-1, and IL-6, in human THP-1 monocytes [104], suggesting that the fucoidan content in the 5F/1C nanoparticle plays an important role in inducing anti-inflammatory effects in human THP-1 monocytes. These outcomes allow us to envisage the possibility of fucoidan being used as an anti-inflammatory agent in the treatment of inflammatory diseases, especially when developed as a nanoparticle.

All these results further strengthen the concept that fucoidan may induce its anti-inflammatory effects by inhibiting pro-inflammatory cytokine secretion and mRNA expression levels of pro-inflammatory cytokines (TNF-α, IL-1β, and IL-6) in LPS-stimulated immune cells including RAW264.7 murine macrophages [96,97,99,101,102,108], Caco-2 cells [107], THP-1 cells [105], human PBMC, porcine peripheral blood PMNs, and MSCs. More importantly, macrophages appear to play an essential role in the initial stages of the anti-inflammatory effects of fucoidan. However, Obluchinskaya et al. (2022) demonstrated that *Fucus vesiculosus* fucoidan extracts (FV1 and FV3) containing a high polyphenol content (135.9 PhE/g DW) induced a greater radical scavenging activity than four other seaweeds (*Saccharina japonica*, *Fucus distichus*, *Fucus serratus*, and *Ascophyllum nodosum*) [17], suggesting that the chemical composition in fucoidans may also influence its capability to induce anti-inflammatory effects.

Fucoidan antioxidant effects have been shown to prevent disruptions caused by the excessive accumulation of amyloid-β and reactive oxygen species (ROS) in the human body [109]. Several previous in vitro studies have demonstrated that fucoidans can act as ROS scavengers by removing free hydroxyl radicals and superoxide radicals [110,111]. The antithrombotic and anticoagulant effects of fucoidans are demonstrated primarily by prolonged activated partial thromboplastic time (APTT), prothrombin time (PT), and thrombin time (TT) [60], indicating that fucoidans can exhibit inhibitory effects on several intrinsic factors (e.g., II, V, VIII, IX, XI, XII) and extrinsic pathways to induce antithrombotic and anticoagulant effects in in vitro studies [62,109]. Fucoidans have also been shown to help relieve symptoms of various viral diseases, including severe acute respiratory syndrome coronavirus 2 (SRAS-CoV-2) [75], I-type influenza virus [73], human immunodeficiency virus 1 (HIV-1) [70], and herpes simplex virus (HSV) [72], by inhibiting virus attachment to host cells or directly inhibiting specific viral-related antigen productions. Fucoidans have also been shown to inhibit angiogenesis by regulating vascular endothelial growth factors (VEGF), fibroblast growth factor 2 (FGF-2), matrix metalloproteinases (MMPs), and chemokines (e.g., CXCL12) [67,109].

### 2.3. Pharmacokinetics of Fucoidans

The absorption of fucoidans is dependent on their source and structure. Several studies have investigated fucoidan absorption using the ELISA method and suggested that the molecular weight of a fucoidan may influence its absorption and excretion [112]. However, this observation is not entirely consistent with those obtained from other relevant in vivo and human clinical studies [113,114,115,116,117,118,119,120]. They suggested that the dosage, species, and structure of a fucoidan would also contribute to the variation in absorption and elimination [113,114,115,116,117]. Zhao et al. reported that oral administered LMWF (*Laminaria japonica*) reached the highest concentration in rat plasma at 15 h [113], which was faster than LMWF (*Laminaria japonica*, MW = 35 kDa) at the same oral dose [113]. The results suggest that the molecular weight of fucoidan below 10 kDa with a high oral dose would help to increase the absorbed amount of fucoidan in a relatively short timeframe (15 h vs. 25 h). It also indicated that the molecular weight of fucoidan ranging from 10–40 kDa has a long absorption time in rat plasma. Others showed that oral administration of LMWF (*Laminaria japonica*) in rat plasma reached the $C_{\max}$ in under 2 h [114,115], which has a shorter absorption time after oral administration. When Zhao et al. (2016) used a higher oral dose of fucoidan (oral dose = 400 mg/kg and 800 mg/kg) in rats, the time to take to reach $C_{\max}$ was much longer. Thus, the oral dose of LMWF could also be a critical factor affecting the pharmacokinetics of fucoidan in rats. Furthermore, Pozharitskaya et al. demonstrated that the fucoidan concentration (*Fucus vesiculosus*), in the plasma of rats reached a maximum level (0.125 μg/g) at 4 h after intragastric administration. These results were not entirely consistent with those obtained from human clinical studies. A human study showed that the plasma concentration of fucoidan (*Cladosiphon okamuranus*, molecular weight = 66 kDa, dose = 1 g) reached the maximum level in 7 out of 10 samples (serum: up to 100 ng/mL) at 6 h after 1 g of fucoidan administered to healthy volunteers, and only one sample reached the highest concentration (approximately 62 ng/mL) at 9 h [116]. These results indicate that the human intestinal absorption rate of fucoidan varies between individuals. They also found that the concentration of fucoidan was elevated in urine (up to 1000 ng/mL) after oral administration [116], but the molecular weight of fucoidan was reduced to 1.8–3.1 kDa in urine [116]. These results indicate that orally administered LMWF has a higher absorption and elimination rate than intragastrical administered HMWF, suggesting that the molecular weight of fucoidan could be a critical factor influencing the pharmacokinetics of fucoidan in the body. Fucoidan species is less likely to be a factor affecting the pharmacokinetics of fucoidan, suggesting that LMWF from *Fucus vesiculosus* or *Cladosiphon okamuranus* is capable of being developed and used for therapeutic purposes.

The pharmacokinetics of fucoidan is also influenced by the route of administration. A LMWF (*Laminaria japonica*,) reached the $C_{\max}$ (110.53 μg/mL) in 5 min after I.V. injection [118], but oral administered LMWF (*Laminaria japonica*,) was detected at 2 h after intragastric administration [118]. Similar results were also obtained from a recent study by Bai et al. (2020); they demonstrated that the fluorescein isothiocyanate labeled (FITC) fucoidan (*Fucus vesiculosus*, MW = 107.8 kDa, I.V. injection dose = 50 mg/kg) reached the $C_{\max}$ (66.37 μg/g) in mouse plasma at 30 min and was not detectable in the blood after 4 h [120]. They also showed that FITC fucoidan could circulate to other organs in mice, such as the lung ($C_{\max}$ = 110.92 μg/g at 4 h), liver ($C_{\max}$ = 284.7 μg/g at 0.5 h), spleen ($C_{\max}$ = 77.79 μg/g at 6 h), and kidney ($C_{\max}$ = 1092.31 μg/g at 4 h) [120]. Data on pharmacokinetics in humans is still limited.

### 2.4. Biomedical Usages of Fucoidans

Fucoidans are important pharmaceutical candidates in cancer therapy because of their high nutritional and biomedical value. Fucoidans have been classified as a dietary supplement that is generally recognized as safe by the FDA and is recognized as a safe food ingredient [42,121,122]. Prior to becoming an FDA-approved food supplement in 2017 and commercially available on the market, seaweed has been widely used as a functional food and as an established ingredient in local cuisine in several Asian countries, including China, Japan, and Korea [55]. During the 16th to 18th centuries, seaweed was discovered to have medicinal benefits in the treatment of goiter, psoriasis, asthma, several thyroid deficiencies, and skin diseases in China, France, and the United States [10]. In addition, fucoidan derived from *Fucus vesiculosus* has been used in a variety of combination products in European countries, including (1) homeopathic medical products in Austria; (2) laxatives in Belgium and Poland; (3) an authorized iodine supplement in Denmark; (4) an adjuvant in slimming diets in France, Spain, and the United Kingdom, and (5) as a traditional herbal remedy to treat obesity and rheumatic pain in the United Kingdom [10]. Fucoidan, as a nutritional supplement, is administered primarily through oral tablets and liquid administration. On the other hand, fucoidan is currently used as an ingredient in cosmetic and nutraceutical products.

## 3. Immunopotentiating Effects of Fucoidans

Fucoidan was first observed to support the immune response in the early 1980s [123]. The effect of fucoidans on immune potentiation emerged in the 1990s [124,125]. Over the past two decades, research into the immunomodulatory (immunopotentiating) effects of fucoidans has expanded significantly.

### 3.1. Overall Effects of Fucoidans on the Immune System

Fucoidans have now been shown to have immunopotentiating effects on both the adaptive and innate immune response (Figure 1). The immunopotentiating effects of fucoidan have been identified in studies using T cells [2,126,127], macrophages [2,36,128], DC (DCs) [94,127], and natural killer cells (NK) [72,127,129,130]. Toll-like receptors (TLRs) and scavenger receptor type A (SR-A) receptors are the two critical controllers in fucoidan-activated DCs [35,131,132]. Pro-inflammatory cytokines, such as TNF-α, IFN-γ, and interleukin 6 (IL-6), have been proposed as a mechanism to regulate fucoidan-activated macrophages and NK cells to induce antitumor immunity dependent on dose and molecular weight [35,95,133].

### 3.2. Immunological Effect of Fucoidans on T Cells

Fucoidan immunomodulatory effects are preferentially studied using T cells, particularly when investigating its antitumor immunity. However, studies investigating the direct effect of fucoidans on T cells are relatively limited. So far, there has been no study examining the effect of fucoidan on CAR-T cells.

Fucoidan antitumor immune responses have been studied in both in vivo and in vitro models [4,94,132,134]. When co-cultured CD8+ T cells and human breast cancer cells (MCF-7) were used to investigate the immune response of fucoidan, it was discovered that the number of CD8+ T cells and interferon-γ (IFN-γ) was doubled in the fucoidan-treated group compared to the control [93]. However, when NY-ESO-1-specific CD8+ T cells were stimulated by fucoidan (*Fucus vesiculosus*)-treated DCs, interferon-γ (IFN-γ) secretion was higher than in the CD8+ T cells group [93], suggesting that fucoidan can activate NY-ESO-1-specific CD8 + T cells by increasing IFN-γ production. Similar results were also obtained from a separate study, which demonstrated that fucoidan promotes both CD4+ and CD8+ T cell responses by upregulating the pro-inflammatory cytokines of Th1 and Tc1 cells, namely IFN-γ and TNF-α [94]. However, because this immune response depends on IL-12 production, fucoidan may be able to enhance Th1 and Tc1 immune responses if it increases IL-12 production in the presence of Th1 and Tc1 cells. Furthermore, this study also demonstrated that fucoidan could be used as an adjuvant to enhance T cell responses by upregulating the production of IFN-γ, as well as increasing CD44+ CD4 cell and memory T cell proliferation [94]. These findings suggest that fucoidan induces direct immune responses in CD4 and CD8 T cells acting as an adjuvant to boost T cell immune responses.

Yang et al. demonstrated that fucoidans from *Ascophyllum nodosum* and *Fucus vesiculosus* directly promote T cell proliferation and activation through upregulating IFN-γ and TNF-α secretion in CD8+ T cell populations [18]. Furthermore, gene set enrichment analysis (GSEA) revealed that some representative genes within the JAK/STAT pathway were increased in the fucoidan treatment group, including IL-3, IL-6, IL-13, IL-14, IL-24a, CSF2, and CD70 indicating that fucoidan improves CD8+ T cell activation and proliferation via the JAK/STAT pathway [18]. More importantly, they found that the T cell receptor (TCR) complex played a critical role in promoting the activation and proliferation of CD8+ T cells and that it interacted with the TCR/CD3 complex to enhance T cell activation [18], which has a significant advantage in the discovery of new molecular mechanisms of fucoidan-induced immunomodulatory effects. These results are partially consistent with another study in which the treatment with intranasal fucoidan (*Ecklonia cava*) treatment increased the level of IFN-γ and TNF-α in mLN CD8 and CD4 T cells in both C57BL/6 and BALB/c mice [23]. This study explored a new route of administration of fucoidan in mice whilst also showing that fucoidan (*Ecklonia cava)* can activate T cells and enhance T cell proliferation by increasing the production levels of IFN-γ and TNF-α [23]. Lee et al. showed that *Undaria pinnatifida* fucoidan-rich extract significantly increased CD4+ and CD8+ T cells proliferation in cyclophosphamide (CP)-treated immunosuppressed mice via increasing the expression levels of cytokines (IFN-γ, TNF-α) and plasma antibodies (TgM, total IgG) [135]. These results suggest that IFN-γ and TNF-α are the two critical effectors involved in fucoidan-induced immunological effects in CD4+ and CD8+ T cells, specifically in T cell activation and proliferation. In particular, this new discovery regarding the different effects of fucoidan administration needs to be further investigated, especially its immunomodulatory effects on T cells using the dominant fucoidan species, *Fucus vesiculosus*. Combined, these experimental results demonstrate that fucoidan has potent immunomodulatory effects on T cells.

Fucoidan can also induce indirect immune responses in T cells. LMWF (*Undaria pinnatifida*) has been shown to induce indirect immune responses in CD4+ and CD8+ T cells through DCs [35], as DCs treated with LMWF activated T cells and significantly increased CD4+ and CD8+ T cell proliferation, indicating that DCs treated with LMWF play a critical role in T cell activation and proliferation.

Table 2 summarizes the immune cells and cytokines involved in the mechanism of immunopotentiation of fucoidans.

### 3.3. Immunological Effect of Fucoidan on Dendritic Cells

In early studies, DCs were considered to be a potential target for the immunomodulatory capacity of fucoidan [131]. The DCs are critical mediator cells that interact with fucoidan to activate pattern recognition receptors (PRR), resulting in T cell priming and acquired immunity [139]. Toll-like receptors (TLRs) and scavenger receptors (SRs) have emerged as critical regulators that can interact with DC and fucoidan to influence innate and adaptive immunity [94,139,140]. Fucoidan can activate DC maturation by increasing specific surface molecules, such as CD40, CD86, MHC class I and class II, or cytokines, including IL-12 [94], TNF-α [132] and IFN-γ, in the host [131].

#### 3.3.1. Fucoidan Activates the Maturation of DCs via Toll-like Receptors

The TLRs are essential receptors for initiating DCs and triggering primary immune responses and are typically involved in the recognition of potential pathogens and the activation of DCs [139]. Activation of TLRs can alter the ability of DCs to interact with T cells by regulating three types of signals delivered by DC to promote T cell expansion and differentiation into effectors: (1) antigen-specific signal 1; (2) co-stimulatory ‘signal 2′ proteins, such as CD40, CD80, and CD 86; (3) cytokines, such as TNF-α and INF-γ [139,141]. As previously reported, LMWF from *Undaria pinnatifida* stimulated the maturation of DCs by activating the toll-like receptor 4 (TLR4), and its downstream MAPK and NF-κB signaling pathways markedly increasing the expression of CD40, CD86, MHC I, and MHC II molecules [35].

Another LMWF fucoidan (*Ecklonia cava*) has also been shown to promote similar activation of DCs by increasing the levels of CD40, CD80, CD86, and MHC class I and II molecules in fucoidan-treated mice, upregulating the production of several pro-inflammatory cytokines, such as IL-6, IL-12, and TNF-α [130]. In this study, the LMWF-mediated DC maturation process is activated through the TLR4 signaling pathway. Mechanistically, fucoidan activated the TLR4 signaling pathway by upregulating the phosphorylation level of ERK, JNK, p38, and p-iκB while downregulating the level of p-NF-κB p65 [35]. This discovery supports the role of TLR4 as a critical mediator in fucoidan-activated DC maturation and cytokine production.

Fucoidan (*Fucus vesiculosus*) also contributed to activation of the maturation of human monocyte-derived DCs [131], as well as the upregulation of the expression of co-stimulatory molecules of DCs, promoting the secretion of cytokines, namely TNF-α, IL-12 and IFN-γ [131].

It is worth noting that different sources of fucoidan used in the above studies could lead to the activation of DCs, implying that the actual source of fucoidan may not be a critical factor. One common feature of LMWFs was that in all experiments they were shown to be endotoxin-free, which differ from HMWFs that potentially were contaminated with endotoxin.

#### 3.3.2. Fucoidans Activate the Maturation of DCs via Scavenger Receptor Type A (SR-A)

Fucoidans inhibited SR-A in DCs by significantly increasing the binding of NY-ESO-1 to DCs. The DCs treated with fucoidan (species data not shown) exhibited a more mature phenotype than the DCs without fucoidan treatment [93]. This result indicates that SR-A is a critical factor that may influence DC maturation. Fucoidan (*Fucus evanescens*) was shown to indirectly induce human peripheral blood dendritic cell maturation (PBDC) and increase TNF-*α* production [132]. Therefore, fucoidan (*Fucus evanescens*)-medicated DC maturation can be blocked if fucoidan is pretreated with a TNF-α-neutralizing antibody [132]. Furthermore, a consistent result was obtained in a later in vivo study, in which they not only demonstrated that fucoidan (*Fucus vesiculosus*) could induce maturation in mouse DCs and induce upregulation of TNF-α, but also showed that fucoidan (*Fucus vesiculosus*) increased the production of IL-6 and IL-12 in spleen DCs [94]. These findings imply that fucoidan is a viable candidate for induced antitumor immune responses by regulating the levels of IFN-γ, TNF-α, or other specific interleukins, such as IL-6 and IL-12 levels.

### 3.4. Immunological Effect of Fucoidan on Macrophages

The immunomodulatory effect of fucoidan via macrophages is strongly associated with the production of pro- and anti-inflammatory cytokines, such as interleukin 6 (IL-6) [36,95], TNF-α [36,95,128], nitric oxide (NO) [128,137], and inducible nitric oxide synthase (iNOS) [95,128,142].

In the innate and adaptive immune systems of mammals, macrophages are the first line of defense against pathogens and tumors [95]. Macrophages have been shown to interact with fucoidan to inhibit tumor initiation, progression, and metastasis within the malignant tumor microenvironment (TME) [133,143]. A recent study demonstrated that LMWF from *Undaria pinnatifida* markedly increased the number of macrophages in splenocytes with elevated levels of IL-6 secretion [35], implying that fucoidan-mediated IL-6 secretion may play an essential role in macrophage activation.

Fucoidan has been shown to induce immunomodulatory effects after being treated with M2 macrophages by suppressing cytokines (IL-6 and TNF-α), and downregulating CCL22 chemokine by inhibiting p65-NF-κB phosphorylation [95,133]. However, different results were found in a different study where they demonstrated that LMWF-treated RAW264.7 macrophages increased the expression of pro-inflammatory mediators of IL-6 and TNF-α [95], as well as enhancing NO and iNOS production [95]. Furthermore, LMWF also activated the NF-κB signaling pathway by upregulating the phosphorylation of lκB-α and p65, and upregulated the MAPK signaling pathway by inducing phosphorylation of p38 [95]. Increased NO production in macrophages could be through the p38 MAPK and NF-κB signaling pathway [137]. A fractionated fucoidan (*Nizamuddinia zanardiinii*) named as $F_{3}$, could activate RAW264.7 macrophages by increasing the expression of cytokine mRNA (iNOS, NO, TNF-α, IIL-1β, and IL-6) and proteins that regulate the MAPK and NF-κB signaling pathway, such as p-NF-κB, p-JNK, p-ERK, and p-p38 proteins [128]. In summary, these results show that fucoidan activates RAW264.7 macrophages by upregulating the inflammatory cytokine, MAPK, and NF-κB signaling pathways.

Furthermore, fucoidan (*Laminaria japonica*) has been used to promote the differentiation of RAW264.7 macrophages from M0 to M1 phenotype macrophages [36]. Differentiation of RAW264.7n macrophages to M1 macrophages occurs via increasing the pro-inflammatory cytokines of IL-6, TNF-α, and NO [36]. Fucoidan compounds KCA (*Kjellmaniiella crassifolia*, *Astragalus polysaccharide*, and *Codonopsis pilosula*) and UCA (*Undaria pinnatifida*, *Astragalus polysaccharide*, and *Codonopsis pilosula*) may increase macrophage proliferation by increasing GM-CSF and TNF-α at concentrations below 200 μm/mL (KCA: 50–100 μm/mL; UCA: 25–200 μm/mL) [144]. However, when the concentrations of fucoidan compounds were above 200 μm/mL, fucoidan killed the RAW264.7 macrophages. Surprisingly, fucoidan from a few sources (for example, *Kjellmaniiella crassifolia* and *Undaria pinnatifida* fucoidan) showed different effects, such as inhibiting the growth of macrophages opposed to cell death [144].

### 3.5. Immunological Effect of Fucoidan on NK Cells

The NK cells are innate lymphoid cells that are essentially derived from common lymphoid progenitors [145]. The broad range of cytokines (e.g., IFN-γ, perforin, granzyme B) [127,130,145], and activating (e.g., NKp30, FasL) and inhibitory receptors (e.g., killer inhibitory receptor, KIR) located on the cell surface [145,146] allow them to interact with other immune cells or biomolecules which allow them to recognize tumor cells and then induce antitumor effects [145,147]. Furthermore, NK cells have also been described as DC promoters and T cell response regulators [147], suggesting that NK cells can enhance and maximize the antitumor effects of other immune cells in TME [145]. Ale et al. showed that intraperitoneally administered fucoidan (*Fucus vesiculosus*, 50 mg/kg) increased NK cell proliferation in C57BL/6 mice [129]. Fucoidan from *Fucus vesiculosus* and *Undaria pinnatifida* could significantly increase NK cells ($\mathrm{NK}1.1^{+}\mathrm{CD}3^{-}$) proliferation. Fucoidan from other sources (*Fucus vesiculosus*, *Undaria pinnatifida*, *Ascophyllum nodosum*, *Macrocystis pyrifera*) could activate NK cells ($\mathrm{CD}3^{-}\mathrm{NK}1.1^{+})$ via increasing killer cell lectin-like receptor and IFN-γ [127]. Furthermore, Zhang et al. (2019) further demonstrated that intraperitoneal administration of fucoidan (*Macrocystis pyrifera*, 50 mg/kg) in C57BL/6 mice could activate and increase NK cell proliferation of NK cells ($\mathrm{NK}1.1^{+}\mathrm{CD}3^{-}$) via increasing IFN-γ production and CD69 expression [148], suggesting that fucoidan activates NK cells through secreted cytokines (e.g., IFN-γ) and killer activating receptors (e.g., KLRG1). Exceptionally, $\mathrm{CD}3^{-}\mathrm{NK}1.1^{+}$ NK cells activated by all fucoidans from *Fucus vesiculosus*, *Undaria pinnatifida*, *Ascophyllum nodosum*, and *Macrocystis pyrifera* induced cytotoxic activity against YAC-1 cells [127]. Importantly, both Zhang et al. (2015) and Zhang et al. (2019) showed that fucoidan (*Macrocystis pyrifera*, 50 mg/kg)-activated $\mathrm{CD}3^{-}\mathrm{NK}1.1^{+}$ NK cells induced the highest cytotoxicity in mouse lymphoma YAC-1 cells [127,148], which suggest that fucoidan-activated $\mathrm{CD}3^{-}\mathrm{NK}1.1^{+}$ NK cells could induce antitumor effects, regardless of the fucoidan species. Consistent results were demonstrated in a recent study by Zhang et al. (2021); they showed that intraperitoneal administration of fucoidan (*Ecklonia cava*) in C57BL/6 mice induced the strongest $\mathrm{CD}3^{-}\mathrm{NK}1.1^{+}$ NK cell proliferation through increased CD69 expression and IFN-γ levels among the five fucoidans (*Fucus vesiculosus*, *Undaria pinnatifida*, *Ascophyllum nodosum*, *Macrocystis pyrifera*, and *Ecklonia cava*) at a concentration of 50 mg/kg [130,130]. Fucoidan (*Ecklonia cava*)-treated NK cells also induced strong antitumor effects in YAC-1 cells by significantly upregulating the expression of TRAIL, perforin, and granzyme B on the surface of fucoidan (*Ecklonia cava*)-treated NK cells [130]. Furthermore, an in vivo study by An et al. (2022) demonstrated that fucoidan (*Laminaria japonica*) could promote $\mathrm{CD}3^{-}\mathrm{NK}1.1^{+}$NK cells proliferation via increasing CD69 at a higher concentration of 100 mg/kg and killing targeted cells by secreting IFN-γ, perforin, and granzyme B at a lower concentration of 50 mg/kg [149]. Two other immunological mediators were also involved in the activation and cytotoxicity of fucoidan-mediated NK cells against other tumor cells in vitro, the death ligand FasL and the activating receptor NKp30 [146]. These data suggest that fucoidan would increase NK cells proliferation via increasing CD69 expression and IFN-γ levels and would induce cytotoxic effects of $\mathrm{CD}3^{-}\mathrm{NK}1.1^{+}$ NK cells in YAC-1 cells by upregulating surface markers, including perforin, granzyme B, NKp30, FasL, TRAIL, and KLRG1. Thus, cytotoxic mediators of IFN-γ, namely perforin and granzyme B, play a critical role for fucoidan in the promotion and activation of NK cells in mice.

Fucoidans could also induce cytotoxicity and the activation of NK cells in cyclophosphamide (CP)-treated immunosuppressed mice. It was reported that fucoidan (*Undaria pinnatifida*, 50 mg/kg, 100 mg/kg, 150 mg/kg) induced cytotoxicity against YAC-1 cells and increased $\mathrm{NK}1.1^{+}$NK cells proliferation in CP-treated immunosuppressed mice [135], but it is unknown whether fucoidan secreted IFN-γ or promoted CD69 expression on the surface of $\mathrm{NK}1.1^{+}$NK cells [135]. The HMWF (*Undaria pinnatifida*) markedly increased the proliferation of NK-92MI cells in the concentration range of 62.5 to 2000 μg/mL and induced a high cytotoxicity of NK-92MI cells against YAC-1 cells in CP-treated immunosuppressed mice by releasing granzyme B [49]. These results suggest that fucoidan could promote NK cell proliferation and activation by releasing granzyme B in CP-treated immunosuppressed mice. Oral administration of fucoidans (*Cladosiphon okamuranus*, HMWF: 110–138 kDa, LMWF: 6.5–40 kDa) has significantly increased the proliferation of NK cells in the spleen and reduced tumor weight in mice with tumors in the colon 26 [150], suggesting that fucoidan induces antitumor immunity effects by mediating NK cell activity [150].

It is noteworthy that it was previously suggested that “fucoidan-mediated NK cell activation depends on DC maturation” by showing that fucoidan (*Ecklonia cava*) was unable to increase CD69, IFN-γ, perforin, and granzyme B levels in NK cells after splenocytes depleted $\mathrm{CD}11c^{+}$DCs [130,149]. However, uronic acid levels in fucoidans (*Fucus vesiculosus*, *Undaria pinnatifida*, *Ascophyllum nodosum*, *Macrocystis pyrifera*, and *Ecklonia cava*) have also been suggested to influence the immunological effects of fucoidan in NK cells [130]. Thus, further investigations are required to determine whether the depletion of DCs or the levels of uronic acid influence fucoidan to activate NK cells.

### 3.6. Factors Influencing Fucoidan-Activated Immune Cells

The dosage and molecular weight of fucoidan play a critical role in modulating the effect on macrophages. Indeed, LMWF fucoidan (*Undaria pinnatifida*) significantly enhances macrophage proliferation (CD11b +) and NK cells (CD3-CD19-CD49b+) by upregulating IL-6 secretion [35]. Furthermore, LMWF can restore cyclophosphamide (CTX)-induced immunosuppression without causing toxic effects. This natural multifunctional molecule is capable of alleviating CTX toxicity and acts as an immunomodulator in vivo [35]. These findings lay the basis for future studies to determine whether this optimal dose is appropriate for testing *Fucus vesiculosus* fucoidan using other immune cells, such as CD4+ and CD8+ T cells. Importantly, their findings establish the foundation for future researchers to select appropriate dose ranges that can be used in human clinical studies to examine their antitumor immunity in cancer patients and a suitable route of administration. However, it remains controversial whether the molecular weight of fucoidan influences its immune modulation effects on macrophages [144].

Furthermore, HMWF is also important in inducing immunomodulatory effects on macrophages. To determine whether molecular weight could be a factor that influences fucoidan to induce immunomodulatory effects on macrophages, Jiang et al. reported that HMWF-treated spleen cells, which include macrophages, enhanced the production of IFN-γ and NO production compared to LMWF treatment [138], suggesting that HMWF is more capable than LMWF of stimulating IFN-γ and NO production, increasing macrophage viability. However, increased NO production may be the result of contaminated HMWF used in the study; concern has been raised about endotoxin contamination in HMWF used in studies [95]. To overcome this concern, it is suggested that endotoxin-free LMWF is a better choice for future studies to investigate its immunomodulatory effects of fucoidan [35].

## 4. Relevance to Cancer Immunotherapy

Immunotherapy, i.e., immune checkpoint inhibitors and adoptive T cell transfer therapy (such as CAR-T cell therapy), has achieved an unprecedented clinical outcome in the treatment of homological malignancies. However, significant challenges remain ahead in the treatment of solid tumors. Effective treatment of solid tumor requires preferential activation of CTL-mediated cytotoxicity against cancer cells. To overcome these seemingly intractable obstacles, more ‘powerful’ immune cells or CAR-T cells with enhanced antitumor efficacy are required [20]. Fucoidan is a natural polysaccharide that has antitumor properties and immunopotentiating effects with low toxicity in both animal models and humans. This polysaccharide is a strong agonist of TLR4, which induces activation of DCs in humans. Therefore, fucoidan could be a desirable candidate for use in effective cancer immunotherapy. Figure 2 summarizes the immunopotentiation of fucoidan and its relevance to cancer immunotherapy.

### 4.1. Fucoidan Potentiates Cancer Immunotherapy

There is considerable evidence that fucoidan potentiates cancer immunotherapy when used in combination with immunotherapeutic products. A recent study has shown that fucoidan supplements significantly enhance the antitumor activities of PD-1 antibodies in vivo. Fucoidan consistently promotes the activation of tumor infiltrating CD8+ T cells. Therefore, fucoidan combined with ICB therapy could be a promising strategy to treat cancer. The underlying mechanism for this enhanced immunotherapy may be related to the activation of the JAK/STAT pathway to stimulate T cell activation [18]. Fucoidan (*Ecklonia cava*) combined with anti-PD-ligand 1 (anti-PD-L1) antibody treatment (administered intranasally) successfully prolonged survival for mice with metastatic lung cancer, and the body weight of the treated mice also increased. Furthermore, the combination treatment inhibits the infiltration of B16 tumor cells into the lung and prevents the growth of metastatic murine colon carcinoma CT-26 tumors in the lung [23]. When fucoidan was used together with Nivolumab, it increased the effects of Nivolumab on prostate cancer cells by enhancing the activity of human immune cells, and an additive effect was observed [136]. In a murine tumor model, a combination of fucoidan and anti-PD-L1 antibody inhibited CT-26 tumor growth more effectively than the anti-PD-L1 antibody. The study confirmed that the combination treatment with fucoidan enhanced anti-PD-L1 antibody-mediated anti-cancer immunity in the CT-26 carcinoma-bearing BALB/c mice, providing direct in vivo evidence [21].

### 4.2. Fucoidan Enhances the Efficacy of Immunotherapy via Novel Dosage Forms

Nanotechnology has emerged as a new approach to improve the efficiency of cancer immunotherapy by targeting the drug delivery system and the metabolism of immunotherapeutic agents [151]. Two preclinical studies have shown that fucoidans (*Fucus vesiculosus* and *Laminaria japonica*) have the potential to be used to improve the efficacy of immunotherapy in cancer treatments [152,153]. When fucoidan (*Fucus*
*vesiculosus*) is packed as a therapeutic nanomedicine, it is capable of augmenting the therapeutic index of combination checkpoint immunotherapy and reducing toxicity: Chiang et al. reported the beneficial therapeutic effects of fucoidan–dextran-based magnetic nanomedicine (IO@ FuDex3) conjugated with a checkpoint inhibitor (anti-PD-L1) and T cell activators (anti-CD3 and anti-CD28). The IO@FuDex3 medicine helped the repair of the immunosuppressive tumor microenvironment and potentiated the effect of the anti-PD-L1 antibody [126]. Fucoidan (*Fucus*
*vesiculosus*)-based IL-2 delivery microcapsules, FPC2, loaded with IL-2 showed higher biological activity in ex vivo expansion of cytotoxic T cells than from Treg lymphocytes. A single intratumor administration of the FPC2/IL-2 complex with injectable gel had a favorable effect on the subpopulation ratio of tumor-infiltrating leukocytes as a result of the increased expansion of cytotoxic T lymphocytes and the decreased number of myeloid subpopulations, leading to increased activation of tumor-reactive T cells only at the tumor site. This represents a novel strategy in TCR-engineered T cell therapies for solid tumors [154].

More details on studies using fucoidans in novel dosage forms, such as nanoparticles and microcapsules, to improve immunotherapy efficacy, can be found in the following table (Table 3).

### 4.3. Fucoidan Enhances T Cell Activation via Increasing IL-2, IFN-γ, TNF-α

Several cytokines, including interleukin 2 (IL-2), interleukin 6 (IL-6), interleukin 12 (IL-12), TNF-α, and IFN-γ, have been shown to interact with fucoidan to enhance T cell activation. Here, IL-2 is a pro-inflammatory cytokine usually secreted by activated CD4+ and CD8+ T cells [155]. It is a known T cell growth factor responsible for T cell (CD4+, CD8+) expansion, and the production of TNF-α and IFN-γ [155]. Tomori et al. (2019) showed that the oral administration of fucoidan (*Cladosiphon okamuranus*) could significantly increase the secretion of IL-2 and IFN-γ in spleen cells (T and B cells) in mice, whilst reducing the secretion of IL-4, IL-5, and the serum antibody IgE [138]. They also showed that addition of concanavalin (Con A) in fucoidan-treated mice promotes T cell proliferation [2]. Increased production of pro-inflammatory cytokines (IL-2, IFN-γ) led to proliferation of T cells. Fucoidan administered with PD-1 antibodies induced synergistic effects in inhibiting B16 melanoma tumor cell proliferation, via increasing T cell infiltration to B16 tumor cells [18]. Fucoidan activated the JAK/STAT signaling pathway and augmented CD8+ T cell proliferation via interacting with the TCR/CD3 complex and increasing the production of TNF-α and IFN-γ [18].

These results suggest that fucoidan enhances antitumor immune responses in TME via regulating T cell activities. Another in vivo study by Lee et al. (2020) also demonstrated a similar trend; oral administration of fucoidan (*Undaria pinnatifida*) in cyclophosphamide (CP)-treated C57BL/6 mice significantly increases the proliferation of CD4+ and CD8+ T cells via increasing the production of TNF-α and IgM, at concentrations of 100 mg/kg and 150 mg/kg, respectively [135]. The results suggest that the production of TNF-α and IFN-γ plays a critical role in regulating fucoidan-mediated T cell activations.

More importantly, a very recent in vivo study conducted by Jeon et al. (2022) showed that a single intratumoral injection of fucoidan-based coacervate named as ${\mathrm{FPC}}^{2}$– IG-IL-2 releases a higher level of IL-2 expression and increases the proliferation of CD4+ and CD8+ T cells [154]. They also showed that ${\mathrm{FPC}}^{2}$– IG-IL-2 could enhance the antitumor effectiveness of a PD-1 blockade in both CT26-beraing mice and tumor-infiltrating CD8+ T cells [154]. These results confirmed that ${\mathrm{FPC}}^{2}$– IG-IL-2 could enhance the activation of transferred tumor-reactive CD8+ T cells and antitumor immune responses. Thus, more preclinical studies are needed to investigate the effects of fucoidans on the production of IL-2 in T cells.

### 4.4. Fucoidan Enhance Immunotherapy through Regulating Cytokines Released by Macrophages and NK Cells

By producing various cytokines and interacting with other immune cells, such as DCs or natural killer (NK) cells, macrophages are involved in initiating, promoting, or, alternatively, inhibiting tumor development [156]. It has also been reported that macrophages can regulate the tumor microenvironment via producing iNOS, NO, or other pro-inflammatory cytokines (e.g., TNF-α and IL-6) [156]. Expression of iNOS in macrophages accelerates NO production, but excessive NO production can contribute to promoting tumor progression and metastasis [157]. Thus, inhibition of aberrant iNOS/NO production may improve the efficacy of cancer immunotherapy [157].

Anti-tumor immunity induced by fucoidan has been demonstrated in a study by Takeda et al. where they showed that fucoidan (*Cladosiphon okamuranus* Tokida)-treated RAW264.7 murine macrophages inhibited Sarcoma 180 (S-180) tumor cell growth in the S and $G_{2}$/M phase of the cell cycle. They also showed that the fucoidan activated RAW264.7 macrophages through increasing the iNOS level and NO production via activating NF-κB signaling pathway, respectively [158]. They also demonstrated that the inhibition of iNOS levels could suppress S-180 tumor cell growth [158].

Jiang et al. (2021) also demonstrated that fucoidan (sea cucumber *Stichopus chloronotus*) activates RAW264.7 macrophages via increasing the production of NO, TNF-α, IL-6, and IL-10, as well as being involved in the activation of the TLR4/2 and NF-κB signaling pathways [159]. These results suggest that fucoidan at low concentrations helps to avoid overexpression of iNOS and NO production in fucoidan-treated RAW264.7 macrophages. Thus, fucoidan is a critical regulator of iNOS expression and NO production in RAW264.7 macrophages.

Furthermore, Tabarsa et al. (2020) confirmed that crude fucoidan (*Nizamuddinia zanardinii*) could increase the proliferation of RAW264.7 macrophages at concentrations of 10–50 μg/mL without cytotoxicity and activate RAW264.7 macrophages by increasing NO production at 50 μg/mL [128]. However, they also showed that fucoidan fraction $F_{3}$ (*Nizamuddinia zanardinii*) upregulated the iNOS expression and NO production, accompanied with releasing several pro-inflammatory cytokines (e.g., TNF-α, IL-1β, IL-6) and IL-10 [128]. The IL-10 produced inhibited the overexpression of TNF-α, IL-1β, and IL-6, exhibiting immunological responses [159]. These results suggest that fucoidan could restrain the overactivation of RAW264.7 macrophages by secreting anti-inflammatory cytokines (e.g., IL-10), which helps to balance the ratio of pro- and anti-inflammatory cytokines in fucoidan activated RAW264.7 macrophages. Tabarsa et al. (2020) demonstrated fucoidan fraction $F_{3}$ (*Nizamuddinia zanardinii*) activated RAW264.7 macrophages increasing the amount of p-NF-κB, p-JNK, p-ERK, and p-p38 proteins [128], which suggests that the fucoidan fraction $F_{3}$ (*Nizamuddinia zanardinii*) activation of RAW264.7 macrophages involves the activation of the NF-κB and MAPK signaling pathways.

Fucoidan also activates NK cells by releasing cytokines (e.g., TNF-α, INF-γ) and activating the NF-B and MAPK signaling pathways [128]. For example, Tabarsa et al. (2020) demonstrated that fucoidan fraction $F_{3}$ (*Nizamuddinia zanardinii*) activates NK-92 cells via increasing the expressions of TNF-α, IFN-γ, granzyme-B, perforin, activating receptor NKG2D, and apoptosis-inducing ligand (FasL) [128]. They also demonstrated that fucoidan fraction $F_{3}$ (*Nizamuddinia zanardinii*) could increase the quantity of p-NF-κB, p-JNK, p-ERK, and p-p38 proteins [128]. Taken together, these results suggest that fucoidan fraction $F_{3}$ (*Nizamuddinia zanardinii*) in NK-92 cells activates NF-κB and MAPK signaling pathways.

Fucoidan is a natural and multifunctional agent that also has promising inhibitory effects on pro-inflammatory cytokines (TNF-α, IL-1β and IL-6) in LPS-treated RAW264.7 murine macrophages [96,97]. A previous in vitro study by Lee et al. (2012) found that three *Ecklonia cava* fucoidan fractions ($F_{1}$, $F_{2}$, $F_{3}$) were non-cytotoxic and reduced the expression of iNOS, NO, COX-2, and pro-inflammatory cytokines (TNF-*α*, IL-1*β*, and IL-6) in RAW 264.7 macrophages treated with LPS at a concentration of 12.5–100 μg/mL [99]. Notably, the $F_{3}$ fraction induced maximum inhibitory effects on these cytokines at a concentration of 100 μg/mL [99]. The in vitro study by Fernando et al. (2017) also showed that the fucoidan fraction (*Chnoospora minima*, MW = 95 kDa) fraction F2,4 was not cytotoxic and could decrease NO production and pro-inflammatory cytokine expression in pro-inflammatory cytokines (TNF-*α*, IL-1*β*, and IL-6) in RAW264.7 macrophages treated with LPS at a concentration of 50 μg/mL [100]. Similar results were also obtained by Ni et al. (2020) in an in vivo study and demonstrated that the fucoidan fraction LJSF4 (*Saccharina japonica*, fucoidan fraction: LJSF4) has no cytotoxicity and significantly inhibited the production of pro-inflammatory cytokines, including TNF-*α*, IL-1*β*, and IL-6 in LPS-treated RAW264.7 macrophages at 25 μg/mL [97]. These fucoidan fractions (e.g., $F_{1}$, $F_{2}$, $F_{3}$, F2,4, LJSF4) contained high level of fucose and sulfate content, suggesting that fucoidan fractions (e.g., $F_{1}$, $F_{2}$, $F_{3}$, F2,4, LJSF4) with the highest fucose and sulfate contents would have a greater ability to regulate and balance cytokine expressions in LPS-treated RAW264.7 macrophages. Therefore, the fucose and sulfate content in fucoidan could be one of the influential factors that affects its ability to induce inhibitory effects on pro-inflammatory cytokines in LPS-treated RAW264.7 macrophages. Jeong et al. also reported that fucoidan (*Fucus vesiculosus*) was not cytotoxic and only attenuated the production of pro-inflammatory cytokines of TNF-*α* and IL-1*β* in LPS-treated RAW264.7 murine macrophages at 100 μg/mL [96]. Ni et al. demonstrated that fucoidan fraction LJSF4 (*Saccharina japonica*, fucoidan fraction: LJSF4) has no cytotoxicity and significantly inhibits the production of pro-inflammatory cytokines, including TNF-*α*, IL-1*β*, and IL-6 in LPS-treated RAW264.7 macrophages at 25 μg/mL [97]. These results strongly suggest that fucoidan can inhibit the secretion and expression of pro-inflammatory cytokines (TNF-*α*, IL-1*β*, and Il-6) in RAW264.7 murine macrophages treated with LPS at a lower concentration (≤100 μg/mL) [96], regardless of their species derivation.

Sanjeewa et al. (2018) reported that the fucoidan (*Sargassum horneri*) fraction f4 inhibited the production of iNOS, NO, and COX-2, and reduced the secretion of pro-inflammatory cytokines (TNF-*α*, IL-6) and the ${\mathrm{PEG}}_{2}$ enzyme in LPS-treated RAW264.7 murine macrophages at concentrations of 25–100 μg/mL (the 50% inhibitory concentration (${\mathrm{IC}}_{50}$) = 87.12 μg/mL) without causing any toxicities [101], but it only slightly inhibited IL-6 production at the same concentration [101]. The f4 fraction also inhibited the phosphorylation of IκB-α and p-I κB-α [101]. In particular, the fucoidan fraction f4 (*Sargassum horneri*) contained lower fucose (36.86%) and sulfate (18.47%) contents [101]. Consistently, crude fucoidan (*Sargassum horneri*) was shown to be non-cytotoxic and could suppress NO production, the secretion of pro-inflammatory cytokines (TNF-α, IL-1*β*), and downregulate the expression levels of the iNOS and COX-2 proteins in macrophages [102]. They also demonstrated that crude fucoidan (*Sargassum horneri*) can inhibit the translocation of NF-*κ*B p50 and p65 to the nucleus and downregulate the phosphorylation of p38 and ERK1/2 proteins in LPS-treated RAW264.7 macrophages at the same concentration [102]. These results suggest that fucoidan induced inhibitory effects on pro-inflammatory cytokines in LPS-treated RAW264.7 macrophages via NF-*κ*B and MAPK signaling pathways.

All these results further strengthen the concept that fucoidan may induce its anti-inflammatory effects by inhibiting pro-inflammatory cytokine secretion and mRNA expression levels of pro-inflammatory cytokines (TNF-α, IL-1β, IL-6) in LPS-stimulated RAW264.7 murine macrophages [96,97,99,101,102,108]. More importantly, macrophages appear to play an essential role in the initial stage of the anti-inflammatory effects of fucoidan. However, the role and contributions of T cells (CD4+ and CD8+) are unknown. Therefore, further investigations are required to determine whether fucoidan-treated T cells (CD4+ and CD8+) are able to attenuate pro-inflammatory cytokines (TNF-α, IL-1β, IL-6).

### 4.5. Possible Combined Use of Fucoidan with Immunotherapeutic Products

Fucoidan extracts from three types of species (*Fucus vesiculosus*, *Undariia pinnatifidat*, *Macrocystis pyrifera*) are capable of independently promoting the activation and proliferation of human peripheral blood mononuclear cells (PBMC) in the presence of a T cell activator (anti-CD3) by increasing IFN-γ production. In these studies, fucoidan derived from *Fucus vesiculosus* was the optimal fucoidan based on the potency of these three different types of fucoidan [136]. Moreover, fucoidan induced antitumor effects by stimulating DC maturation and/or macrophage differentiation. Jin et al. (2014) found that when fucoidan was co-administered with ovalbumin antigen, it stimulated DC maturation and then promoted antigen-specific T cell immune responses [94,160].

Fucoidan (*Fucus vesiculosus*) has been developed as a therapeutic nanomedicine to enhance the efficacy of immune check inhibitors (anti-PD-L1) in immunosuppressive TME [126,161]. It has been shown that the fucoidan-based nanomedicine IO@FuDe$x^{3}$ reduces adverse effects and extends patient survival in cancer treatments [5,126]. Although the development of fucoidan-based therapeutic nanoparticles is still in an early stage [152], it is important and promising to develop fucoidan as a nanotechnology-based cancer immunotherapy drug [162,163].

## 5. Relevant Clinical Trials-Anticancer and Immunomodulation Related Trials

In this section, we provide an overview of relevant clinical trials using fucoidan to treat cancer and modulate immune function that are listed on Clinicaltrials.gov and anzctr.org.au. A list of these trials is presented in Table 4.

Given the promising preclinical findings of experiments using fucoidan as an anticancer agent, some clinical trials have been conducted or are proposed to examine the therapeutic potential of fucoidan in the treatment of various human cancers, such as advanced lung cancer [85], breast cancer (ACTRN12615000673549) (Table 4) [24], metastatic colorectal cancer (NCT04066660) (Table 4) [164], and recurrent colorectal cancer [165]. These clinical trials demonstrate that fucoidan is a safe anticancer agent that can be combined with chemotherapeutic drugs to improve survival time and reduce the adverse effects of chemotherapy in cancer treatments. However, three clinical trials have no results published on the website (clinicaltrials.gov), including NCT04342949, NCT04597476, and NCT03130829 (as of 10 October 2022, Table 4). It is important to be aware that the focus of all registered clinical trials on these websites (clinicaltrials.gov, anzctr.org.au) has expanded their investigations from studying anticancer effects to immunomodulatory effects of fucoidan. For example, four clinical studies focused on investigating how fucoidan regulates biomarkers in the immune system (ACTRN12621000872831, ACTRN12605000021673, ACTRN12616000417482, and ACTRN12611000220965) (Table 4). Other clinical trials have not yet started to recruit participants for their study (Table 4). Therefore, a follow-up on the results of these clinical studies to evaluate the efficacy of fucoidan in cancer would provide more insight into the therapeutic potential of fucoidan.

A previous human clinical study conducted between April 2008 and June 2009 by Ikeguchi et al. [165], found that HMWF (*Cladosiphon okamuranus*) prolonged survival time in the HMWF treatment group and suppressed side effects (e.g., fatigue) of chemotherapy drugs in patients with unresectable and recurrent colorectal cancer [165]. They also demonstrated that HMWF (*Cladosiphon okamuranus*) did not exhibit side effects (e.g., allergic dermatitis) in these patients during the 6 months of fucoidan treatment, and no patient suffered severe toxicity [165]. All patients (n = 20) completed fucoidan treatment safely [165]. The results suggested that HMWF (*Cladosiphon okamuranus*) can protect patients from fatigue of grades 2 and 3 during chemotherapy [165]. In contrast, an open-label non-crossover study assessed the efficacy of co-administration of fucoidan with two hormonal therapies (letrozole and tamoxifen) in patients with breast cancer (ACTRN12615000673549) (Table 4) [24], they showed that the oral administration of fucoidan (*Undaria pinnatifida*) did not influence the plasma levels of the active metabolites of tamoxifen and did not cause significant changes in steady state plasma concentrations of hormonal therapies (letrozole, tamoxifen) in female patients with breast cancer (ACTRN12615000673549) [24]. Furthermore, there is no evidence to show that oral administration of fucoidan (*Undaria pinnatifida*) can cause liver metastases (ACTRN12615000673549) [24]. Tsai et al. (2017) conducted a prospective, randomized, double-blind, controlled human clinical trial in the southern city of Taiwan (NCT04066660) (Table 4) [164]. They found that LMWF (*Sargassum hemiphyllum*) significantly increased the disease control rate (DCR) by approximately 23.6% in the study group compared to the control group (NCT04066660) [164], indicating that fucoidan can manage the status of cancer progression. These results suggest that LMWF (*Sargassum hemiphyllum*) can act as a supplementary agent to induce auxiliary effects and there appears to be an improved overall survival trend in patients with metastatic colorectal cancer (NCT04066660) [164]. Furthermore, Hsu et al. (2018) demonstrated that combination therapy (fucoidan and cisplatin, *n* = 50) increased the survival rate by approximately 50% in the HiQ-fucoidan treatment group (*Laminaria japonica*) as compared to the control group [85]. The results suggest that HiQ-fucoidan (*Laminaria japonica*) can act as a supplementary agent in combination therapy to increase cisplatin absorption to prolong survival time in patients with advanced lung cancer [85]. However, the sample size of the control group was smaller than that of the HiQ-fucoidan group *(Laminaria japonica*) group [85]. Therefore, the precision of clinical results can be compromised. Furthermore, they did not monitor whether HiQ-fucoidan (*Laminaria japonica*) could reduce the side effects of cisplatin (e.g., fatigue) to improve quality of life [85]. Therefore, further human clinical studies are required to randomly select equal numbers of patients in each group and also to monitor whether fucoidan treatment can reduce the side effects of the chemotherapy drug (e.g., cisplatin) in patients with advanced lung cancer.

**Table 4 marinedrugs-21-00128-t004:** List of all registered clinical trials using fucoidan to treat cancer and/or other immune dysfunctional diseases.

Clinical Trial Number	Starting Year (Status)	Description	Results
NCT04342949	2018 (unknown)	A double-blind, randomized, placebo-controlled, parallel study investigated fucoidan’s auxiliary effects in patients with locally advanced rectal cancer who received a combined radio/chemotherapy before surgery. They aimed to observe whether fucoidan can improve the quality of life of these patients receiving the neoadjuvant CCRT.	No
NCT04066660	2019 (Recruiting)	A randomized, double-blind, controlled trial evaluated oligo-fucoidan’s efficacy (500–800 Da) in patients with metastatic colorectal cancer. They have also observed whether fucoidan can improve the quality of life and prolong the overall survival rate of these patients.	Yes [164]
NCT04597476	2020 (Recruiting)	A randomized, double-blind phase II trial that evaluated fucoidan’s clinical effect and safety in patients with stage III/IV head and neck squamous cell carcinoma.	No
NCT03130829	2019 (Withdrawn)	A pilot, randomized, double-blind, multicenter study evaluated whether orally administered oligo-fucoidan can improve the quality of life in patients receiving platinum-based chemotherapy with NSCLC.	No
NCT02875392	2016 (Completed)	A randomized, double-blind, parallel study demonstrated that fucoidan improves the metabolic profiles of patients with non-alcoholic fatty liver disease (NAFLD).	Yes [166]
NCT05437887	2022 (Not yet recruiting)	An open-label, prospective, single group study evaluated the effects of fucoidan on the gut microbiota in the patients of atopic dermatitis before and after fucoidan treatment.	No
NCT05461508	2023 (Recruiting)	An open-label, randomized, parallel study investigates the effects of the combination treatment (fucoidan and Vonoprazan) on Helicobacter Pylori eradication rate and gastrointestinal flora.	No
NCT03422055	2018 (Unknown)	An open-label, single-group phase I study that evaluated the tolerance, biodistribution, and dosimetry of fucoidan radiolabeled by Technetium-99 m in patients.	No
ACTRN126 16000417482	2016 (Completed recruitment)	A randomized, double-blind, placebo-controlled, cross-over phase I/II trial that investigated the measurement, modulation, and estimation of net endogenous non-carbonic acid production using the Australian food database following the administration of alkaline supplements in healthy adults.	N/A
ACTRN126 15000673549	2015 (Completed recruitment)	An open-label, non-randomized, single-group phase IV trial investigating the interaction between two systemic complementary and alternative medicines and standard therapy in patients with active breast cancer malignancy.	Yes [24]
ACTRN126 05000021673	2005 (Recruiting)	A non-randomized, double-blind, parallel phase I/II study that evaluated the effects of natural seaweed fucoidan (GFS) on the modulation of the immune system and the mobilization/release of hematopoietic progenitor stem cells from bone marrow to the peripheral blood.	No
ACTRN1262 1000872831	2021 (Not yet recruiting)	A randomized, double-blind, crossover trial determines whether daily fucoidan supplementation can upregulate immune biomarkers during three weeks of intensified exercise training in healthy, recreationally active adults.	No
ACTRN1260 5000021673	2005 (Recruiting)	A non-randomized, double-blind, parallel phase I/II study that evaluated the effects of natural seaweed fucoidan (GFS) on the modulation of the immune system and the mobilization/release of hematopoietic progenitor stem cells from bone marrow to the peripheral blood.	No

## 6. Concluding Remarks

Fucoidans have a range of immunomodulatory/immunopotentiating effects by acting through DCs, macrophages, NK cells, T cells, and B cells. However, very little is known about their effect on CAR-T cells and other engineered immune cells. Evidence is emerging concerning the benefits of combining fucoidans and immunotherapeutic agents, in particular ICIs and CAR-T cells, for the treatment of cancer. Rationally, fucoidan could play an important role in the new era of cancer immunotherapy through its immunopotentiating effects (Figure 2), anti-inflammatory effects, and antitumor properties. Human clinical studies are limited, and the results of clinical trials are not conclusive and inconsistent with preclinical data, indicating that more clinical studies on fucoidans are necessary. Furthermore, fucoidans may have great potential to be developed as a nanoparticle and used with other immunotherapeutic agents together as a novel strategy to treat cancer.

Despite many beneficial effects and promising future prospects for fucoidan, one of the challenges is the standardization of fucoidan, since not all fucoidans are equal in their pharmacological effects. The molecular structure, molecular weight, chemical composition, and bioactivity of fucoidans differ from species to species [16]. The extraction method is another major factor affecting the structural composition and bioactive properties of fucoidans. It is hoped that these problems will be gradually overcome as fucoidan moves forward toward more clinical applications.

## Figures and Tables

**Figure 1 marinedrugs-21-00128-f001:**
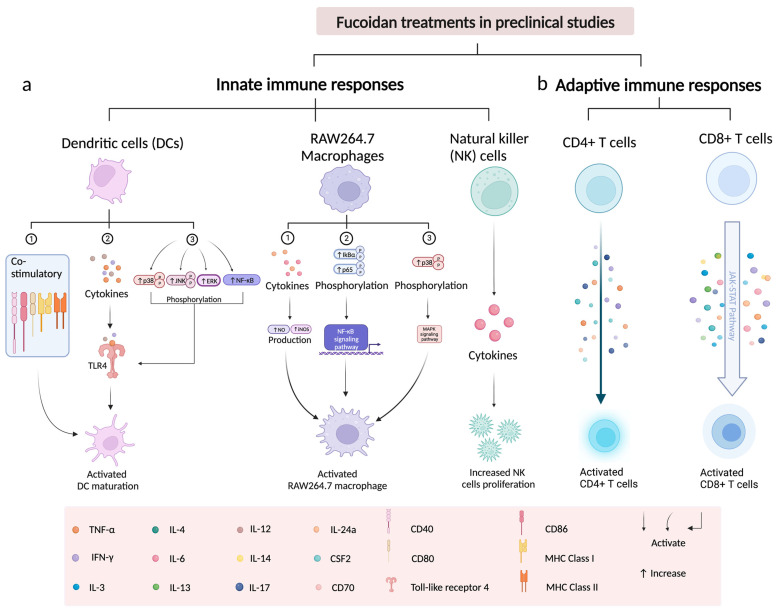
Effects of fucoidans on different types of immune cells. (**a**) Effect of fucoidans on innate immune responses through DC, RAW264.7 murine macrophages, and NK cells; (**b**) effect of fucoidans on adaptive immune responses via CD4+ and CD8+ T cells (this figure was created with BioRender.com, accessed on 5 February 2023).

**Figure 2 marinedrugs-21-00128-f002:**
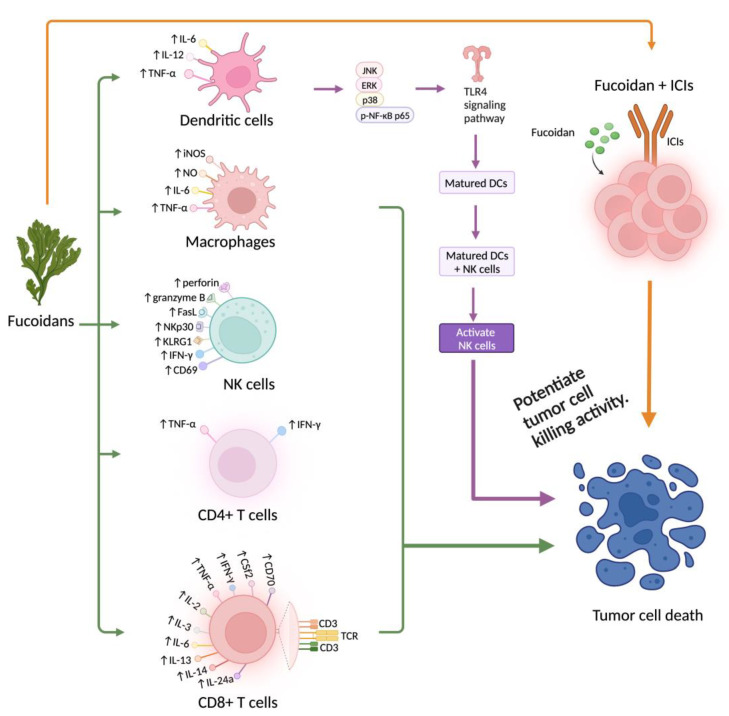
Immunopotentiation and antitumor immunity of fucoidan. Fucoidan activates DC maturation via secreting IL-6, IL-12, and TNF-*α*; matured DCs will then activate NK cells to induce antitumor immunity effects in tumor cells. Fucoidan can activate macrophages via increasing the production of iNOS, NO, IL-6, and TNF-*α*. Fucoidan increases NK cell proliferation by increasing CD69 expression and IFN-*γ* levels and activates NK cells by upregulating the cytotoxic mediators of perforin, granzyme B, activating the receptor NKp30, FasL, and KLRG1. Here, TNF-*α* and IFN-*γ* are the two common cytokines expressed on the surface of CD4+ and CD8+ T cells. Fucoidan activates these T cells via upregulating cytokines of TNF-*α*, IFN-*γ*, CD70, CSf2, IL-2, IL-3, IL-6, IL-13, IL-14, IL-24a, to induce tumor cell death. Fucoidan is co-administered with immune checkpoint inhibitors (ICIs) to enhance tumoricidal activity (this figure was created with BioRender.com, accessed on 5 February 2023).

**Table 2 marinedrugs-21-00128-t002:** The immunomodulatory effects of fucoidans on different types of immune cells.

Fucoidan Sources	Immune Cells	Involved Cytokines	Research Methods	Effects of Fucoidan	Ref.
*Fucus* *vesiculosus*	CD4+ T cells, CD8+ T cells, dendritic cells	IFN-*γ*, TNF-*α*, IL-12, IL-6, IL-12p40	In vivo	Upregulated the production of IFN-*γ* and TNF-*α* in the presence of Th1 and Tc1 cells, to promotes CD8+ and CD4+ T cell responses. Induced maturation of DCs by upregulating TNF-*α*, IL-6, and IL-12p40 in spleen DCs. Increased the cell proliferation of CD44+ CD4 and memory T cells. Acts as an adjuvant to boost T cell immune responses.	[94]
*Fucus* *Vesiculosus*, *Ascophyllum nodosum*	CD4+ T cells, CD8+ T cells	IFN-*γ*, TNF-*α*, IL-3, IL-6, IL13, IL-14, L-24a, CSF2, CD70	In vivo, In vitro	Promoted T cell proliferation and activation through upregulating IFN-*γ* and TNF-*α* secretion in CD8+ T cell populations. Promoted T cell proliferation via the JAK/STAT pathway. Interacted with TCR/CD3 complexes to enhance T cell activation. Co-administration with PD-1 antibody reduced tumor size and weight in mice. Co-administration with PD-1 antibody increased the ratio of CD8+ and CD4+ T cells in the spleen.	[18]
*Fucus* *vesiculosus*	CD8+ T cells, dendritic cells	IFN-*γ*	In vivo	Increased the cell proliferation of CD8+ T cells and also upregulated the production of IFN-*γ*. Inhibited SR-A in DCs by increasing the binding of NY-ESO-1 to DCs. In co-culture fucoidan-treated DCs with CD8+ T cells, serum IFN-*γ* increased.	[93]
*Ecklonia cava*	CD8+ T cells, CD4+ T cells	IFN-*γ*, TNF-*α*	In vivo	Activated T cells and enhanced T cell proliferations by increasing serum IFN-*γ* and TNF-*α*. Co-administration with PD-1 antibody prolonged survival in metastatic lung cancer. Acted as an adjuvant to enhance the therapeutic efficacy of immunotherapy.	[23]
*Undaria* *pinnatifida*	CD8+ T cells, CD4+ T cells, dendritic cells, CD11b+ macrophages, CD3-CD19-CD49b+ NK cells	TNF-*α*, IL-12, IL-6, TLR4, CD40, CD86, MHC I, MHC II, ERK, JNK, p38, p-i*κ*B, p-NF-*κ*B p65	In vivo, In vitro	The LMWF-treated DCs activated T cells. The LMWF-treated DCs significantly increased CD4+ and CD8+ T cell proliferation. Activated the maturation of DCs by upregulating TLR4, CD40, CD86, MHC I, and MHC II. Activated the maturation of DCs by activating the TLR4, MAPK, and NF-*κ*B signaling pathways. Activated the TLR4 signaling pathway by upregulating the phosphorylation of ERK, JNK, p38, and p-i*κ*B while downregulating the level of p-NF-*κ*B p65. Enhanced CD11b+ macrophage and CD3-CD19-CD49b+ NK cell proliferation by upregulating IL-6. Restored CTX-induced immunosuppression.	[35]
*Fucus* *vesiculosus*	Dendritic cells	IL-12, NF-*α*, FN-*γ*	In vitro	Activated the maturation of DCs by upregulating TNF-*α*, IFN-*γ*, and IL-12.	[131]
*Laminaria* *Japonica*, *Laminaria cichoriodes*, *Fucus* *evanescens*	HEK293 (human embryonic kidney cells)	NF-*κ*B, TLR2, TLR4	In vitro	Activated the NF-*κ*B by interacting with human TLR2 and TLR4.	[134]
*Fucus* *vesiculosus*, *Undariia* *pinnatifidat*, *Macrocystis pyrifera*	Human peripheral blood mononuclear cells (PBMCs)	IFN-*γ*	In vitro	Promoted activation and proliferation of PBMCs. Reached the highest PBMC activation by increasing maximum IFN-*γ* secretion. Increased IFN-*γ* secretion after co-culture of Nivolumab-treated PBMCs and PC3 cells in the presence of anti-CD3. Fucoidan reached the highest activation level of PBMCs through increasing maximum IFN-*γ* secretion at the concentration of 10 μg/mL and 50 μg/mL. Inhibited PC3 proliferation.	[136]
*Fucus* *evanescens*	Human peripheral blood dendritic cells	TNF-*α*	In vivo	Induced PBDC maturation and increased TNF-*α*.	[132]
*Fucus* *vesiculosus*	M2 macrophages	TNF-*α*, IL-6, CCL22	In vitro	Inhibited TNF-*α* and IL-6. Downregulated CCL22 chemokine by inhibiting p65-NF-*κ*B phosphorylation.	[133]
*Undaria* *pinnatifida*	RAW 264.7 macrophages	TNF-*α*, IL-6, NO, iNOS, p38, *κ*B-*α*, p65	In vitro	Increased the expression of IL-6 and TNF-*α*. Enhanced NO and iNOS production. Activated the NF-*κ*B signaling pathway by upregulating the phosphorylation levels of l*κ*B-*α* and p65. Activated the MAPK signaling pathway by increasing p38 phosphorylation.	[95]
Unknown	RAW 264.7 macrophages	iNOS, NO, p38, SR-A	In vitro	Activated iNOS and increased NO production through the NF-*κ*B and MAPK signaling pathways.	[137]
*Undaria* *pinnatifida sporophyllus*, *Fucus* *vesiculosus*	Spleen cells, B lymphocytes	IFN-*γ*, NO, CD25, CD69	In vitro	Enhanced spleen cell proliferation and viability. Increased the expression of CD19, IFN-*γ* and NO on spleen cells. Increased a higher expression level of CD25 and CD69 on B lymphocytes. Increased the level of CD25 and CD69 on B lymphocytes. Increased spleen cell proliferation and viability. Reduced necrotic spleen cell populations.	[138]

**Table 3 marinedrugs-21-00128-t003:** Fucoidans in novel dosage form enhance immunotherapy efficacy.

Source of Fucoidan	Brief Description	Dosage Forms	Effects of Fucoidan	Ref.
*Fucus vesiculosus*	Combined fucoidan-based magnetic nanoparticles and immunomodulators enhance tumor-localized immunotherapy.	Nanoparticles (fucoidan-containing formulations: IO@FuDex1, IO@FuDex2, IO@FuDex3, M-IO@ FuDex1, M-IO@ FuDex3, and M-IO@ FuDex3)	IO@Fu${\mathrm{Dex}}^{2}\;$-H: improved targeting efficiency; IO@Fu${\mathrm{Dex}}^{2}$ and IO@Fu${\mathrm{Dex}}^{3}$: increased the cell association via a slow elevation of median fluorescence index (MFI) in 4T1 cells; IO@Fu${\mathrm{Dex}}^{3}$-H: significantly increased the MFI in $\mathrm{CD}8^{+}$T cells; targeted PD-L1 receptors and associated with 4T1 cells; inhibited lung metastasis in 4T1 cancer model; M-IOFuDex (magnetic navigation): enhanced tumor selectivity; increased T cell proliferation; decreased Tregs and TAMs in TME; IO@FuDex and IO@Fu${\mathrm{Dex}}^{3}$: inhibited the CT-26 tumor cell growth and extended the median survival to 62 days; reduced spleen Tregs; IO@Fu${\mathrm{Dex}}^{3}$ and M-IO@Fu${\mathrm{Dex}}^{3}$ increased TNF-α, VEGF, and TGF-β; IO@Fu${\mathrm{Dex}}^{1}$, IO@Fu${\mathrm{Dex}}^{2}$, IO@Fu${\mathrm{Dex}}^{3}$, and M-IO@Fu${\mathrm{Dex}}^{3}:\;↑$ antitumoral effects and median survival.	[126]
*Fucus vesiculosus*	Enhanced adoptive T cell therapy using fucoidan-based IL-2 delivery microcapsules.	Microcapsules (fucoidan-based coacervate-laden injectable hydrogel (${\mathrm{FPC}}^{2}-\mathrm{IG}$))	Acted as an IL-2 delivery vehicle for enhancing adoptive T cell therapy (ACT); increased tumor-infiltrating $\mathrm{CD}8^{+}$ T cells in CT26-bearing mice with ${\mathrm{FPC}}^{2}-\mathrm{IG}-\mathrm{IL}-2$ injection than ${\mathrm{FPC}}^{2}-\mathrm{IG}$ injection; Downregulated CD62L and enriched $T_{\mathrm{EM}}$ and $T_{\mathrm{EFF}}$ cell generation; promoted STAT5 phosphorylation in $\mathrm{CD}8^{+}$ T cells; increased Treg, NK, DNT, NKT, B, $\mathrm{CD}8^{+}$, and $\mathrm{CD}4^{+}$ T cell populations; induced a higher Ki-67 expression in CT26-bearing mice; combination therapy (anti-PD-1 + ${\mathrm{FPC}}^{2}-\mathrm{IG}-\mathrm{IL}-2$) reduced CT26 tumor cell growth and increased the IFN-γ levels in tumor-infiltrating $\mathrm{CD}8^{+}$T cells; increased naïve OT-I T and NY-ESO-1 TCR- T cell proliferation; decreased the expression of PD-1, Tim-3, TIGIT, and LAG-3 in tumor-infiltrating NY-ESO-1 TCR T cells.	[154]
*Fucus vesiculosus*	Cytotoxicity and fabrication of fucoidan-cisplatin nanoparticles for macrophage and tumor cells.	Nanoparticles	Increased the cell viability of RAW264.7 macrophages; non-cytotoxic to RAW264.7 macrophages; reduced the cytotoxicity of cisplatin; inhibited HCT-8 cell growth.	[152]
*Laminaria japonica*	Fucoidan-based and tumor-activated nanoplatform overcame hypoxia and enhanced photodynamic therapy and antitumor immunity.	Nanoparticles	Significantly increased the VP fluorescent emission in FM@VP-treated MDA-MB-231 cells; MDA-MB-231 cells took up greater FM@VP nanoparticle clusters; inhibited MDA-MB-231 and MDA-MB-468 cell growth; decreased TNBC cell viability, upregulated P-selectin level; overcome tumor hypoxia; decreased pro-angiogenesis generated by hypoxic tumor-elicited pro-angiogenesis; inhibited YAP levels, CTGF, cyclin D1, and EGFR in MDA-MB-231 cells; attenuated the Hippo signaling; downregulated the protein levels of PD-L1; enhanced T cell-mediated cytotoxicity; suppressed orthotopic 4T1 tumor cells growth and metastatic colonization of lung tumor; downregulated Treg cell infiltration; increased the expressions of granzyme B and IFN-γ; increased CD4 and CD8 T cells but decreased TAMs.	[153]
*Cladosiphon* *okamuranus*	Immunomodulatory effects of fucoidan in mice.	Oral gavage	Increased the proliferation of splenocytes that activated by concanavalin A and LPS, and increased macrophage phagocytosis activity and the levels of IL-2, IFN-γ and serum IgM; decreased the levels of IL-4, IL-5 and serum IgE.	[2]
*Fucus vesiculosus*, *Ascophyllu nodosum*	Fucoidan-supplemented diet coordinated with ICBs to potentiate its antitumor immunity	Oral	Enhanced the therapeutic efficacy of PD-1 blockade; reduced B16 melanoma cell growth, volumes, and weights. Increased the proliferation of $\mathrm{CD}8^{+}$T, NK, and tumor-infiltrating T cells; activated DC maturation; increased the proliferation of $\mathrm{CD}8^{+}$T cells via increasing the production of IFN-γ and TNF-α; activated $\mathrm{CD}8^{+}$T cells through the JAK/STAT pathway.	[18]

## Data Availability

The data (figures and tables) used to support the findings of this study are included within the article.

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
