# Peer review of "Immunopotentiating Activity of Fucoidans and Relevance to Cancer Immunotherapy"

_marinedrugs, 2023, doi:10.3390/md21020128_

Round 1

Reviewer 1 Report

I have read the manuscript and I have some questions and comments.
1. The manuscript does not contain a "Materials and Methods" section. Therefore, it is not clear how it differs from many others published recently on the topic "Fucoidan" (for example, https://doi.org/10.3390/md20070412 and others). Please include a "Materials and Methods" section indicating the purpose of this manuscript, years of search, keywords, databases, etc.
2. Figure 1 is not correct. The work (https://doi.org/10.3390/md20100638) presents an idealized scheme of fucoidan. Please correct this figure or delete.

3. The manuscript is lacking of important details. Without these details, it seems that the authors collected information not from full articles, but from abstracts. Therefore, please include in section 2.1 the table with the details on fucoidans from various algae, indicating the name of the algae, place of collection, extraction method, yield, monosaccharide composition, molecular weight, literature reference. Please discuss this information.
4. Section 2.2 is also not informative. No data in numbers, no discussion. Please include in section 2.2 a table with in vivo/in vitro data indicating the object of study, test model, positive control, results in numbers, literature reference. Discuss the effect of molecular weight on different activities. For example, high molecular weight fucoidan showed a significant effect (doi: 10.3390/md18050275). Fucoidan from different sources has different effects (https://doi.org/10.3390/md20100606).
5. The manuscript has no information on the topical use of fucoidan. Recent work shows the preference for external use of fucoidan over oral (https://doi.org/10.3390/md17120687, https://doi.org/10.3390/md19110643). Please complete the manuscript with the necessary information. Discuss the possibility of using fucoidan as an effective anti-inflammatory agent.
6. Table 1 must be supplemented with experimental data in numbers.
7. Section 4.2 contains no information. Please complete this section with a table including information: source of fucoidan, its brief description, type of dosage form, effects, reference.

Author Response

Please see attached PDF file for details.

Reviewer 2 Report

The authors did a comprehensive review on fucoidans and their Immunopotentiating activity. The text is very attractive to readers. The subjects are well distributed. However, there are some issues that need to be corrected before the manuscript can be considered for publication.

On page 2, line 88. Authors make it clear that the structure of a fucoidan depends on several factors. I agree. But, I would like this information to be clear in the introduction as well. Because in this topic, the authors are always using the word “fucoidan” and they don't talk about fucoidans, or say what fucoidan they are talking about. This conveys to the reader that there is only one fucoidan, and that regardless of the source it was extracted from, all fucoidan are the same. And that's not true. In line 75 the authors even distinguish fucoidan, but not in the rest of the introduction.

Still in the introduction, on page 1, lines 45 and 46. The authors state that fucoidans are found in organisms other than algae. Authors need to review this. It seems that there was a confusion among the authors, and they started to consider fucans as fucoidans. There are papers in the literature that call fucans as fucoidans, but this is a mistake and needs to be corrected. I advise you to read Glycobiology. 2003 Jun;13(6):29R-40R. doi: 10.1093/glycob/cwg058. Also, confirm if there is a record of fucoidans in seagress, as the references (5 and 6) cited by the authors do not confirm this.

Topic 2.1 should be improved. The authors try to review the structure of fucoidan, but only present data on fucoidan from the alga Fucus vesiculosus. And the worst is that they present a wrong structure (fig. 1) of this fucoidan. It is important that the authors talk about the structure of other fucoidan, especially the other algae mentioned in the manuscript: Laminaria japonica, Cladosiphon okamuranus, Undaria pinnatifida, etc.

In topic 3.3.1 the authors speak about the action of fucoidan from Undaria pinnatifida. Then they talk about the fucoidan of Ecklonia cava, and finally about Fucus vesiculosus. The text thus makes it clear to the reader that there are different fucoidans and that they have different mechanisms of action.

However, there are many topics where the data are not presented like this, that is, the authors do not make it clear that fucoidan has such activity. As in topic 2.2. A reader viewing this thread will believe that all fucoidans have antitumor activity, have antioxidant, anti-inflammatory and immunomodulatory activity, etc.

Attributing the activity of one fucoidan to all other fucoidans is an unacceptable mistake. And the authors have done this in various parts of their manuscript. Therefore, the authors need to correct this, and present the data as presented in topic 3.3.1

Author Response

Thank you for your constructive comments.

Please see attached PDF file for details.

Round 2

Reviewer 1 Report

I have read the revised manuscript. The authors made the necessary corrections. I have no more questions.

Reviewer 2 Report

The authors improved the manuscript. I now feel confident in saying that the forthcoming paper will bring a lot of attention from readers. Therefore, I am in favor of publishing this manuscript.